# Relation-First Modeling Paradigm for Causal Representation Learning toward the Development of AGI

## Abstract

The traditional i.i.d.-based learning paradigm faces inherent challenges in addressing causal relationships, which has become increasingly evident with the rise of applications in causal representation learning. Our understanding of causality naturally requires a perspective as the creator rather than observer, as the "what...if" questions only hold within the possible world we conceive. The traditional perspective limits capturing dynamic causal outcomes and leads to compensatory efforts such as the reliance on hidden confounders. This paper lays the groundwork for the new perspective, which enables the *relation-first* modeling paradigm for causality. Also, it introduces the Relation-Indexed Representation Learning (RIRL) as a practical implementation, supported by experiments that validate its efficacy.

## 1 Introduction

The concept of Artificial General Intelligence (AGI) has prompted extensive discussions over the years yet remains hypothetical, without a practical definition in the context of computer engineering. The pivotal question lies in whether human-like "understanding", especially causal reasoning, can be implemented using formalized languages in computer systems Newell (2007); Pavlick (2023); Marcus (2020). From an epistemological standpoint, abstract entities (i.e., perceptions, beliefs, desires, etc.) are prevalent and integral to human intelligence. However, in the symbol-grounded modeling processes, variables are typically assigned as observables, representing tangible objects to ensure their values have clear meaning.

Epistemological thinking is often anchored in objective entities, seeking an irreducible "independent reality" Eberhardt & Lee (2022). This approach necessitates a metaphysical commitment to constructing knowledge by assuming the unproven prior existence of the "essence of things", fundamentally driven by our desire for certainty. Unlike physical science, which is concerned with deciphering natural laws, technology focuses on devising effective methods for problem-solving, aiming for the optimal functional value between the nature of things and human needs. This paper advocates for a shift in perspective when considering technological or engineering issues related to AI or AGI, moving from traditional epistemologies to that of the creator. That is, our fundamental thinking should move from "truth and reality" to "creation and possibility".

In some respects, both classical statistics and modern machine learnings traditionally rely on epistemology and follow an "object-first" modeling paradigm, as illustrated by the practice of assigning pre-specified, unchanging values to variables regardless of the model chosen. In short, individual *objects* (i.e., variables and outcomes) are defined a priori before considering the *relations* (i.e., model functions) between them by assuming that what we observe precisely represents the "objective truth" as we understand it. This approach, however, poses a fundamental dilemma when dealing with causal relationship models.

Specifically, "causality" suggests a range of possible worlds, encompassing all potential futures, whereas "observations" identify the single possibility that has actualized into history with 100% certainty. Hence, addressing causal questions requires us to adopt the perspective of the "creator" (rather than the "observer"), to expand the objects of our consciousness from given entities (i.e., the observational world) to include possible worlds, where values are assigned "as supposed to be", that is, *as dictated by the relationship*.

Admittedly, causal inference and related machine learning methods have made significant contributions to knowledge developments in various fields Wood (2015); Vuković (2022); Ombadi et al. (2020). However, the

inherent misalignment between the "object-first" modeling principle and our instinctive "relation-first" causal understanding has been increasingly accentuated by the application of AI techniques, i.e., the neural network-based methods. Particularly, integrating causal DAGs (Directed Acyclic Graphs), which represent established knowledge, into network architectures Marwala (2015); Lachapelle et al. (2019) is a logical approach to efficiently modeling causations with complex structures. However, surprisingly, this integration has not yet achieved general success Luo et al. (2020); Ma et al. (2018).

As Scholkopf Schölkopf et al. (2021) points out, it is commonly presumed that "the causal variables are given". In response, they introduce the concept of "causal representation" to actively construct variable values as causally dictated, replacing the passively assumed observational values. However, the practical framework for modeling causality, especially in contrast to mere correlations, remains underexplored. Moreover, this shift in perspective suggests that we are not just dealing with "a new method" but rather a new learning paradigm, necessitating in-depth philosophical discussions. Also, the potential transformative implications of this "relation-first" paradigm for AI development warrant careful consideration.

This paper will thoroughly explore the "relation-first" paradigm in Section 2, and introduce a complete framework for causality modeling by adopting the "creator's" perspective in Section 3. In Section 4, we will propose the *Relation-Indexed Representation Learning* (RIRL) method as the initial implementation of this new paradigm, along with extensive experiments to validate RIRL's effectiveness in Section 5.

## 2 Relation-First Paradigm

The "do-calculus" format in causal inference Pearl (2012); Huang (2012) is widely used to differentiate the effects from "observational" data $X$, and "interventional" data $do(X)$ Hoel et al. (2013); Eberhardt & Lee (2022). Specifically, $do(X = x)$ represents an intervention (or action) where the variable $X$ is set to a specific value $x$, distinct from merely observing $X$ taking the value $x$. However, given the causation represented by $X \rightarrow Y$, why doesn't $do(Y = y)$ appear as the action of another variable $Y$?

Particularly, distinct from the independent state $X$, the notation $do(X)$ incorporates its timing dimension to encompass the process of "becoming $X$" as a dynamic. Such incorporation can be applied to any variable, including $do(Y)$, as we can naturally understand a relationship $do(X) \rightarrow do(Y)$. For example, consider the statement "storm lasting for a week causes downstream villages to be drowned by the flood," if $do(X)$ is the storm lasting a week, $do(Y)$ could represent the ensuing water-level enhancement, leading to the disaster.

The challenge of accounting for $do(Y)$ arises from the empirical modeling process. In the observational world, $do(X)$ is associated with clearly observed timestamps, like $do(X_t)$, allowing us to focus on modeling its observational states $X_t$ by treating timing $t$ as a solid reference frame. However, when we conceptualize a "possible world" to envision $do(Y)$, its potential variations can span across the timing dimension. For instance, a disaster might occur earlier or later, with varying degrees of severity, based on different possible conditions. This variability necessitates treating timing as a computational dimension.

However, this does not imply that the timing-dimensional distribution is insignificant for the outcome $Y$. The necessity of incorporating $do(X)$ in modeling highlights the importance of including dynamic features. Specifically, Recurrent Neural Networks (RNNs) are capable of autonomously extracting significant dynamics from sequential observations $x$ to facilitate $do(X) \rightarrow Y$, eliminating the requirement for manual identification of $do(X)$. In contrast, statistical causal inference often demands such identifications Pearl (2012), such as specifying the duration of a disastrous storm on various watersheds under differing hydrological conditions.

In RNNs, $do(X)$ is optimized in latent space as representations related to the outcome $Y$. Initially, they feature the observed sequence $X^t = (X_1, \ldots, X_t)$ with determined timestamps $t$, but as representations rather than observables, they enable the computational flexibility over timing, to assess the significance of the $t$ values or mere the orders. The capability of RNNs to effectively achieve significant $do(X)$ has led to their growing popularity in relationship modeling Xu et al. (2020). However, can the same approach be used to autonomously extract $do(Y)$ over a possible timing?

Since the technique has emerged, facilitating $do(Y)$ is no longer considered a significant technical challenge. It is unstrange that inverse learning has become a popular approach Arora (2021) to compute $do(Y)$ as merely another observed $do(X)$. However, the concept of a "possible world" suggests dynamically interacted

elements, implying a conceptual space for "possible timings" rather than a singular dimension. This requires a shift in perspective from being an "observer" to becoming the "creator". This section will explore the philosophical foundations and mathematically define the proposed *relation-first* modeling paradigm.

## 2.1 Philosophical Foundation

Causal Emergence Hoel et al. (2013); Hoel (2017) marks a significant philosophical advancement in causal relationship understanding. It posits that while causality is often observed at the micro-level, a macro-level perspective can reveal additional information, denoted as Effect Information (EI), such as $EI(X \to Y)$. For instance, consider $Y_1$ and $Y_2$ as two complementary components of $Y$, i.e., $Y = Y_1 + Y_2$. In this case, the macro-causality $X \to Y$ can be decomposed into two micro-causal components $X \to Y_1$ and $X \to Y_2$. However, $EI(X \to Y)$ cannot be fully reconstructed by merely combining $EI(X \to Y_1)$ and $EI(X \to Y_2)$, since their informative interaction $\phi$ cannot be included by micro-causal view, as illustrated in Figure 1(b).

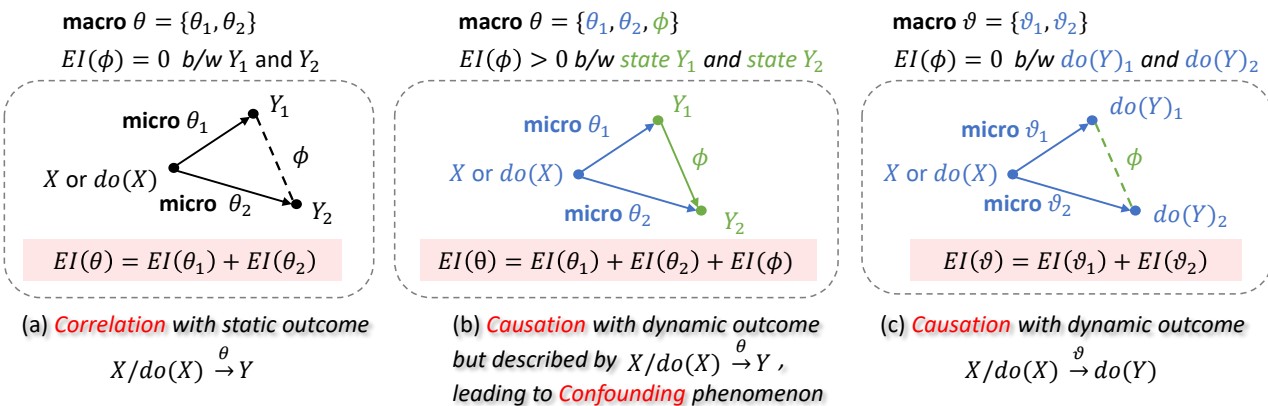

Figure 1: Causal Emergence $EI(\phi) > 0$ stems from overlooking the potential existence of $do(Y)$.

Specifically, the concept of EI is designed to quantify the information generated by the system during the transition from the state of $X$ to the state of $Y$ Tononi & Sporns (2003); Hoel et al. (2013). Furthermore, $\phi$ denotes the minimum EI that can be transferred between $Y_1$ and $Y_2$ Tononi & Sporns (2003). For clearer interpretation, Figure 1(a) illustrates the uninformative statistical dependence between states $Y_1$ and $Y_2$, represented by the dashed line with $EI(\phi) = 0$.

However, this phenomenon can be explained by the information loss when reducing a *dynamic* outcome $do(Y)$ to be a *state* $Y$. Let's simply consider the reduction from $do(X) \to do(Y)$ to $X \to Y$, likened with: attributing the precipitation on a specific date (i.e., the $X_t$ value) solely as the cause for the disastrous high water-level flooding the village on the 7th days (i.e., the $Y_{t+7}$ value), regardless of what happened on the other days. From a computational standpoint, given observables $X \in \mathbb{R}^n$ and $Y \in \mathbb{R}^m$, this reduction implies the information within $\mathbb{R}^{n+1} \cup \mathbb{R}^{m+1}$ must be compactively represented between $\mathbb{R}^n$ and $\mathbb{R}^m$.

If simplifying the possible timing as the extention of observed timing $t$, identifying a significant $Y_{t+1}$ can still be feasible. However, since $Y_1 \to Y_2$ implies an interaction in a "possible world", identifying representative value for outcome $Y$ may prove impractical. Suppose $Y_1$ represents the impact of flood-prevention operations, and $Y_2$ signifies the daily water-level "without" these operations. A dynamic outcome $do(Y)_1 + do(Y)_2$ can easily represent "the flood crest expected on the 7th day has been mitigated over following days by our preventions", but it would be challenging to specify a particular day's water rising for $Y_2$ "if without" $Y_1$.

As Hoel (2017) highlights, leveraging information theory in causality questions allows for formulations of the "nonexistent" or "counterfactual" statements. Indeed, the concept of "information" is inherently tied to **relations**, irrespective of the potential **objects** observed as their outcomes. Similar to the employment of the abstract variable $\phi$, we utilize $\theta$ to carry the EI of transitioning from $X_t$ to $Y_{t+7}$. Suppose $\theta$ = "flooding", and $EI(\theta)$ = "what a flooding may imply", we can then easily conceptualize $do(X)$ = "continuous storm" as its cause, and $do(Y)$ = "disastrous water rise" as the result in consciousness, without being notified the specific

precipitation value $X_t$ or a measured water-level $Y_{t+7}$. In other words, our comprehension intrinsically has a "relation-first" manner, unlike the "object-first" approach we typically apply to modeling.

The so-called "possible world" is **created** by our conciousness through innate "relation-first" thinking. In this world, the timing dimension is crucial; without a potential *timing distribution*, "possible observations" would lose their significance. For instance, we might use a model $Y_{t+7} = f(X^t)$ to predict flooding. However, instead of "knowing the exact water level on the 7th day", our true aim is understanding "how the flood might unfold; if not on the 7th day, then what about the 8th, 9th, and so on?" With advanced representation learning techniques, particularly the success of RNNs in computing dynamics for the cause, achieving a dynamic outcome should be straightforward. Inversely, it might be time to reassess our conventional learning paradigm, which is based on an "object-first" approach, misaligned with our innate understanding.

The "object-first" mindset positions humans as observers of the natural world, which is deeply embedded in epistemological philosophy, extending beyond mere computational sciences. Specifically, given that questions of causality originate from our conceptual "creations", addressing these questions necessitates a return to the creator's perspective. This shift allows for the treatment of timing as computable variables rather than fixed observations. Picard-Lindelöf theorem represents time evolution by using a sequence $X^t = (X_1, \dots, X_t)$ like captured through a series of snapshots. The information-theoretic measurements of causality, such as directed information Massey et al. (1990) and transfer entropy Schreiber (2000), have linguistically emphasized the distinction between perceiving $X^t$ as "a sequence of *discrete* states" versus holistically as "a *continuous* process". The introduction of do-calculus Pearl (2012) marks a significant advancement, with the notation $do(X)$ explicitly treating the action of "becoming $X$" as a *dynamic unit*. However, its differential nature let it focus on an "identifiable" sequence $\{\dots, do(X_{t-1}), do(X_t)\}$ rather than the integral $t$-dimension. Also, $do(Y)$ still lacks a foundation for declaration due to the observer's perspective. Even assumed discrete future states with relational constraints defined Hoel et al. (2013); Hoel (2017) still face criticism for an absence of epistemological commitments Eberhardt & Lee (2022).

Without intending to delve into metaphysical debates, this paper aims to emphasize that for technological inquiries, shifting the perspective from that of an epistemologist, i.e., an observer, to that of a creator can yield models that resonate with our instinctive understanding. This can significantly simplify the questions we encounter, especially vital in the context regarding AGI. For purely philosophical discussions, readers are encouraged to explore the "creationology" theory by Mr.Zhao Tingyang.

## 2.2 Mathematical Definition of Relation

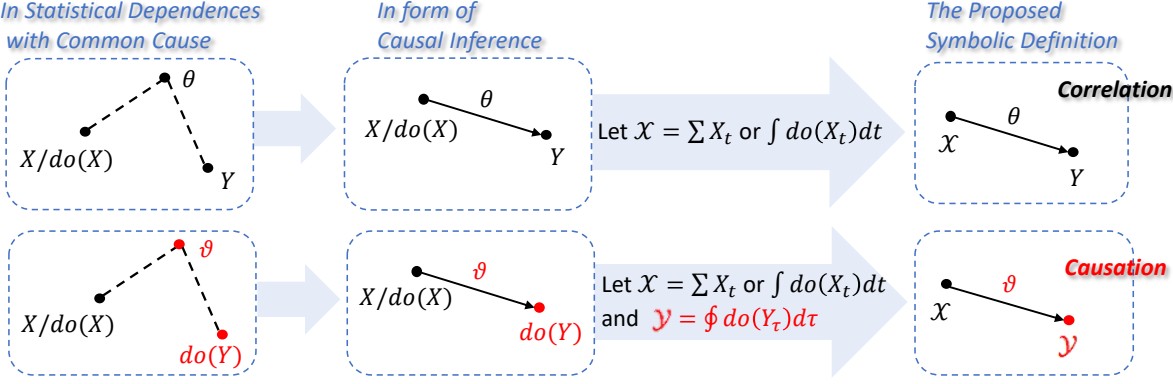

Figure 2: The relation-first symbolic definition of causal relationship versus mere correlation.

A statistical model is typically defined through a function $f(x \mid \theta)$ that represents how a parameter $\theta$ is *functionally related* to potential outcomes $x$ of a random variable $X$ Ly et al. (2017). For instance, the coin flip model is also known as the Bernoulli distribution $f(x \mid \theta) = \theta^x (1-\theta)^{1-x}$ with $x \in \{0, 1\}$, which relates the coin's propensity (i.e. its inherent possibility) $\theta$ to $X = $ "land heads to the potential outcomes". Formally, given a known $\theta$, the *functional relationship* $f$ yields a probability density function (pdf) as $p_\theta(x) = f(x \mid \theta)$, according to which, $X$ is distributed and denoted as $X \sim f(x; \theta)$. The Fisher Information $\mathcal{I}_X(\theta)$ of $X$ about

$\theta$ is defined as $\mathcal{I}_X(\theta) = \int_{\{0,1\}} (\frac{d}{d\theta} log(f(x \mid \theta))^2 p_\theta(x) dx$, with the purpose of building models on the observed $x$ data being to obtain this information. For clarity, we refer to this initial perspective of understanding functional models as the ***relation-first*** *principle*.

In practice, we do not limit all functions to pdfs but often shape them for easier understanding. For instance, let $X^n = (X_1, \ldots, X_n)$ represent an $n$-trial coin flip experiment, while to simplify, instead of considering the random vector $X^n$, we may only record the number of heads as $Y = \sum_{i=1}^n X_i$. If these $n$ random variables are assumed to be independent and identically distributed (i.i.d.), governed by the identical $\theta$, the distribution of $Y$ (known as binomial) that describes how $\theta$ relates to $y$ would be $f(y \mid \theta) = \binom{n}{y} \theta^y (1-\theta)^{n-y}$. In this case, the conditional probability of the raw data, $P(X^n \mid Y = y, \theta) = 1/\binom{n}{y}$ does not depend on $\theta$, implying that once $Y = y$ is given, $X^n$ becomes independent of $\theta$, although $X^n$ and $Y$ each depend on $\theta$ individually. It concludes that no information about $\theta$ remains in $X^n$ once $Y = y$ is observed Fisher et al. (1920); Stigler (1973), denoted as $EI(X^n \rightarrow Y) = 0$ in the context of relationship modeling. However, in the absence of the i.i.d. assumption and by using a vector $\vartheta = (\theta_1, \ldots, \theta_n)$ to represent the propensity in the $n$-trial experiment, we find that $EI(X^n \rightarrow Y) > 0$ with respect to $\vartheta$. Here, we revisit the foundational concept of Fisher Information, represented as $\mathcal{I}_{X \rightarrow Y}(\theta)$, to define:

> **Definition 1.** A relationship denoted as $X \xrightarrow{\theta} Y$ is considered meaningful in the modeling context due to an *informative* ***relation*** $\theta$, where $\mathcal{I}_{X \rightarrow Y}(\theta) > 0$, simplifying as $\mathcal{I}(\theta) > 0$.

Specifically, rather than confining within a function $f(; \theta)$ as its parameter, we treat $\theta$ as an individual variable to encapsulate the effective information (EI) as outlined by Hoel. Consequently, the *relation-first principle* asserts that a relationship is characterized and identified by a specific $\theta$, regardless of the appearance of its outcome $Y$, leading to the following inferences:

1. $\mathcal{I}(\theta)$ inherently precedes and is independent of any observations of the outcome, as well as the chosen function $f$ used to describe the outcome distribution $Y \sim f(y; \theta)$.
2. In a relationship identified by $\mathcal{I}(\theta)$, $Y$ is only used to signify its potential outcomes, without any further "observational information" associated with $Y$.
3. In AI modeling contexts, a relationship is represented by $\mathcal{I}(\theta)$; as a latent space feature, it can be stored and reused to produce outcome observations.
4. Just like $Y$ serving as the outcome of $\mathcal{I}(\theta)$, variable $X$ is governed by preceding relational information, manifesting as either observable data $x$ or priorly stored representations in modeling contexts.

**About Relation $\theta$**

As emphasized by the Common Cause principle Dawid (1979), "any nontrivial conditional independence between two observables requires a third, mutual cause" Schölkopf et al. (2021). The crux here, however, is "nontrivial" rather than "cause" itself. For a system involving $X$ and $Y$, if their connection (i.e., the critical conditions without which they will become independent) deserves a particular description, it must represent unobservable information beyond the observable dependencies present in the system. We use $\theta$ as an abstract variable to carry this information $\mathcal{I}(\theta)$, unnecessarily referring to tangible entities.

Traditionally, descriptions of relationships are constrained by objective notations and focus on "observable states at specific times". For instance, to represent a particular EI, a state-to-state transition probability matrix $S$ is required Hoel et al. (2013). But $S$ is not solely sufficient to define a $EI(S)$, which also accounts for how the current state $s_0 = S$ is related to the probability distributions of past and future states, $S_P$ and $S_F$, respectively. More importantly, manual specification from observed time sequences is necessitated to identify $S_P$, $S$, and $S_F$ irrespective of their observable timestamps. However, the advent of representation learning technology facilitates a shift towards "relational information storage", eliminating the need to specify observable timestamps. This allows for flexible computations across the timing dimension when the resulting observations are required, laying the groundwork for embodying $\mathcal{I}(\theta)$ in modeling contexts.

For an empirical understanding of $\theta$, let's consider an example: A sociological study explores interpersonal ties using consumption data. Bob and Jim, a father-son duo, consistently spend on craft supplies, indicating

the father's influence on the son's hobbies. However, the "father-son" relational information, represented by $\mathcal{I}(\theta)$, exists solely in our perception - as knowledge - and cannot be directly inferred from the data alone. Traditional *object-first* approaches depend on manually labeled data points to signify the targeted $\mathcal{I}(\theta)$ in our consciousness. In contrast, *relation-first* modeling seeks to derive $\mathcal{I}(\theta)$ beyond mere observations, enabling the autonomous identification of data-point pairs characterized as "father-son".

Since the representation of $\mathcal{I}(\theta)$ is not limited by observational distributions, it allows outcome computation across the timing dimension. This capability is crucial for enabling "causality" in modeling, transcending mere correlational computations. Specifically, we use the notations $\mathcal{X}$ and $\mathcal{Y}$ to indicate the integration of the timing dimension for $X$ and $Y$, and represent a relationship in the general form $\mathcal{X} \xrightarrow{\theta} \mathcal{Y}$. We will first introduce $\mathcal{X}$ as a general variable, followed by discussions about the relational outcome $\mathcal{Y}$.

**About Dynamic Variable $\mathcal{X}$**

> **Definition 2.** For a variable $X \in \mathbb{R}^n$ observed as a time sequence $x^t = (x_1, \ldots, x_t)$, a **dynamic** variable $\mathcal{X} = \langle X, t \rangle \in \mathbb{R}^{n+1}$ is formulated by integrating the *timing* $t$ as an additional dimension.

Time series data analysis is often referred to as being "spatial-temporal" Andrienko et al. (2003). However, in modeling contexts, "spatial" is interpreted broadly and not limited to physical spatial measurements (e.g., geographic coordinates); thus, we prefer the term "observational". Furthermore, to avoid the implication of "short duration" often associated with "temporal," we use "timing" to represent the dimension $t$. Unlike the conventional representation in the sequence $X^t = (X_1, \ldots, X_t)$ with static $t$ values (i.e., the timestamps), we consider $\mathcal{X}$ holistically as a *dynamic* variable, similarly for $\mathcal{Y} = \langle Y, \tau \rangle \in \mathbb{R}^{m+1}$. The probability distributions of $\mathcal{X}$, as well as $\mathcal{Y}$, span both *observational* and *timing* dimensions simultaneously.

Specifically, $\mathcal{X}$ can be viewed as the integral of discrete $X_t$ or continuous $do(X_t)$ over the timing dimension $t$ within a required range. The necessity for representation by $do(X_t)$, as opposed to $X_t$, underscores the **dynamical significance** of $\mathcal{X}$. Put simply, if $\mathcal{X}$ can be formulated as $\mathcal{X} = \sum_1^t X_t$, it equates to $X^t = (X_1, \ldots, X_t)$ in modeling. Conversely, $\mathcal{X} = \int_{-\infty}^{\infty} do(X_t)dt$ portrays $\mathcal{X}$ as a *dynamic*, marked by significant dependencies among $X_{t-1}, X_t$ for unconstrained $t \in (-\infty, \infty)$. Essentially, $do(X_t)$ represents a differential unit of continuous timing distribution over $t$, highlighting not just the observed state $X_t$ but also the significant dependence $P(X_t \mid X_{t-1})$, challenging the i.i.d. assumption. The "state-dependent" and "state-independent" concepts refer to Hoel's discussions in causal emergence Hoel et al. (2013).

> **Theorem 1.** Timing becomes a necessary *computational dimension* if and only if the required variable necessatates *dynamical significance*, characterized by a **nonlinear** *distribution* across timing.

In simpler terms, if a distribution over timing $t$ cannot be adequately represented by a function of the form $x_{t+1} = f(x^t)$, then its nonlinearity is significant to be considered. Here, the time step $[t, t+1]$ is a predetermined constant timespan value. RNN models can effectively extract dynamically significant $\mathcal{X}$ from data sequences $x^t$ to autonomously achieve $\mathcal{X} \xrightarrow{\theta} Y$, due to leveraging the relational constraint by $\mathcal{I}(\theta)$. In other words, RNNs perform indexing through $\theta$ to fulfill dynamical $\mathcal{X}$. Conversely, if "predicting" such an irregularly nonlinear timing-dimensional distribution is crucial, the implication arises that it has been identified as the causal effect of some underlying reason.

**About Dynamic Outcome $\mathcal{Y}$**

> **Theorem 2.** In modeling contexts, identifying a relationship $\mathcal{X} \xrightarrow{\theta} \mathcal{Y}$ as *Causality*, distinct from mere *Correlation*, depends on the **dynamical significance** *of the outcome* $\mathcal{Y}$ as required by $\mathcal{I}(\theta)$.

Figure 2 illustrates the distinction between causality and correlation, where an arrow indicates an informative relation and a dashed line means statistical dependence. If conducting the integral operation for both sides of the do-calculus formation $X/do(X) \to Y$ over timing, we can achieve $\mathcal{X} \to \sum_1^\tau Y_\tau$ with the variable $\mathcal{X}$

allowing to be dynamically significant but the outcome $\sum_1^\tau Y_\tau$ certainly not. Essentially, to guarantee $\mathcal{Y}$ presenting in form of $y^\tau = (y_1, \ldots, y_\tau)$ to match with predetermined timestamps $\{1, \ldots, \tau\}$, do-calculus manually conducts a differentiation operation on the relational information $\mathcal{I}(\theta)$ to discretize the timing outcome. This process is to confirm specific $\tau$ values at which $y_\tau$ can be identified as the effect of a certain $do(x_t)$ or $x_t$. Accordingly, the state value $y_\tau$ will be defined as either the interventional effect $f_V(do(x_t))$ or the observational effect $f_B(x_t)$, with three criteria in place to maintain conditional independence between these two possibilities, given a tangible elemental reason $\Delta\mathcal{I}(\theta)$ (i.e., identifiable $do(x_t) \to y_\tau$ or $x_t \to y_\tau$):

$$\mathcal{Y} = f(\mathcal{X}) = \sum_t f_V(do(x_t)) \cdot f_B(x_t) = \sum_t \begin{cases} f_B(x_t) = y_\tau & \text{with } f_V(do(x_t)) = 1 \text{ (Rule 1)} \\ f_V(do(x_t)) = y_\tau & \text{with } f_B(x_t) = 1 \text{ (Rule 2)} \\ 0 = y_\tau & \text{with } f_V(do(x_t)) = 0 \text{ (Rule 3)} \\ \text{otherwise} & \text{not identifiable} \end{cases} \Bigg\} = \sum_\tau y_\tau$$

In contrast, the proposed *dynamic* notations $\mathcal{X} = \langle X, t \rangle$ and $\mathcal{Y} = \langle Y, \tau \rangle$ offer advantages in two respects. First, the concept of $do(Y_\tau)$ can be introduced with $\tau$ indicating its "possible timing", which is unfounded under the traditional modeling paradigm; and then, by incorporating $t$ and $\tau$ into computations, the need to distinguish between "past and future" has been eliminated.

> **Definition 3.** A ***causality*** characterized by a *dynamically significant outcome* $\mathcal{Y}$ can encompass multiple ***causal components***, represented by $\vartheta = (\vartheta_1, \ldots, \vartheta_T)$. Each $\vartheta_\tau$ with $\tau \in \{1, \ldots, T\}$ identifies a timing dimension $\tau$ to accommodate the corresponding ***outcome component*** $\mathcal{Y}_\tau$.
> The overall outcome is denoted as $\mathcal{Y} = \sum_{\tau=1}^T \mathcal{Y}_\tau = \sum_{\tau=1}^T \int do(Y_\tau) d\tau$, simplifying to $\oint do(Y_\tau) d\tau$.

Definition 3, based on the relation-first principle, uses $\vartheta$ to signify causality. Its distinction from $\theta$ implies that the potential outcome $\mathcal{Y}$ must be dynamically significant. Specifically, within a general relationship, denoted by $\mathcal{X} \xrightarrow{\theta} \mathcal{Y}$, the dynamic outcome $\mathcal{Y}$ only showcases its capability to encompass nonlinear distribution over timing, whereas $\mathcal{X} \xrightarrow{\vartheta} \mathcal{Y}$ confirms such nature of this relationship, as required by $\mathcal{I}(\vartheta)$.

According to Theorem 1, incorporating the possible timing dimension $\tau$ when computing $\mathcal{Y}$ is necessary for a causality identified by $\mathcal{I}(\vartheta)$. If a relationship model can be formulated as $f(\mathcal{X}) = Y^\tau = (Y_1, \ldots, Y_\tau)$, it is equal to applying the independent state-outcome model $f(\mathcal{X}) = Y$ for $\tau$ times in sequence. In other words, $\mathcal{X} \xrightarrow{\theta} Y$ is sufficient to represent this relationship without needing $\tau$. It often goes unnoticed that a sequence variable $X^t = (X_1, \ldots, X_t)$ in modeling does not imply the $t$-dimension has been incorporated, where $t$ serves as constants, lacking computational flexibility. The same way also applies to $Y^\tau$.

However, once including the "possible timing" $\tau$ with computable values, it becomes necessary to account for the potential components of $\mathcal{Y}$, which are possible to unfold their dynamics over their own timing separately. For a simpler understanding, let's revisit the example of "storm causes flooding." Suppose $\mathcal{X}$ represents the storm, and for each watershed, $\vartheta$ encapsulates the effects of $\mathcal{X}$ determined by its unique hydrological conditions. Let $\mathcal{Y}_2$ denote the water levels observed over an extended period, such as the next 30 days, if *without* any flood prevention. Let $\mathcal{Y}_1$ indicate the daily variations in water levels (measured in $\pm$cm to reflect increases or decreases) resulting from flood-prevention efforts. In this case, $\vartheta$ can be considered in two components: $\vartheta = (\vartheta_1, \vartheta_2)$, separately identifying $\tau = 1$ and $\tau = 2$.

Specifically, historical records of disasters without flood prevention could contribute to extracting $\mathcal{I}(\vartheta_2)$, based on which, the $\vartheta_1$ representation can be trained using recent records of flood prevention. Even if their hydrological conditions are not exactly the same, AI can extract such relational difference $(\vartheta_1 - \vartheta_2)$. This is because the capability of computing over timing dimensions empowers AI to extract common relational information from different dynamics. From AI's standpoint, regardless of whether the flood crest naturally occurs on the 7th day or is dispersed over the subsequent 30 days, both $\mathcal{Y}_2$ and $(\mathcal{Y}_1 + \mathcal{Y}_2)$ are linked to $\mathcal{X}$ through the same volume of water introduced by $\mathcal{X}$. In other words, while AI deals with the computations, discerning what qualifies as a "disaster" remains a question for humans.

Conversely, in traditional modeling, $\vartheta$ is often viewed as a common cause of both $\mathcal{X}$ and $\mathcal{Y}$, termed a "confounder", and serves as a predetermined functional parameter before computation. Therefore, if such a parameter is accurately specified to represent $\vartheta_2$, when observations $(\mathcal{Y}_1 + \mathcal{Y}_2)$ imply a varied $\vartheta_1$, it becomes

critical to identify the potential "reason" of such variances. If the underlying knowledge can be found, manual adjustments are naturally necessitated for $(\mathcal{Y}_1 + \mathcal{Y}_2)$ to ensure it performs as being produced by $\vartheta_2$; otherwise, the modeling bias will be attributed to this unknown "reason" represented by the difference $(\vartheta_1 - \vartheta_2)$, named a hidden confounder.

**About Dependence $\phi$ between Causal Components**

As demonstrated in Figure 1, by introducing the dynamic outcome components in (c), the causal emergence phenomenon in (b) can be explained by "overflowed" relational information with $\phi$. Here, $do(Y)_1$ and $do(Y)_2$ act as differentiated $\mathcal{Y}_1$ and $\mathcal{Y}_2$, outcome by $\mathcal{I}(\vartheta_1)$ and $\mathcal{I}(\vartheta_2)$. That is, the *relation-first* principle ensures $\vartheta$ to be informatively separable as $\vartheta_1$ and $\vartheta_2$, leaving $\phi$ simply represent their statistical dependence. However, due to their dynamical significance, $\phi$ may impact the conditional timing distribution across $\tau = 1$ and $\tau = 2$.

> **Theorem 3.** Sequential causal modeling is required, if the ***dependence*** between causal components, represented by $\phi$, has dynamically significant impact on the outcome timing dimension.

The sequential modeling procedure was applied in analyzing the "flooding" example, where training $\vartheta_1$ is conditioned on the established $\vartheta_2$ to ensure the resulting representation is meaningful. Specifically, the directed dependence $\phi$ from $\vartheta_2$ to $\vartheta_1$ requires that the timing-dimensional computations of $\mathcal{Y}_1$ and $\mathcal{Y}_2$ occur sequentially, with $\vartheta_1$ following $\vartheta_2$. Practically, the sequence is determined by the meaningful interaction $\mathcal{I}(\vartheta_1 \mid \vartheta_2)$ or $\mathcal{I}(\vartheta_2 \mid \vartheta_1)$, adapted to the requirements of specific applications.

Suppose the two-step modeling process is $\mathcal{Y}_2 = f_2(\mathcal{X}; \vartheta_2)$ followed by $\mathcal{Y}_1 = f_1(\mathcal{X} \mid \mathcal{Y}_2; \vartheta_1)$. According to the adopted perspective, its information explanation can be notably different. From the creator's perspective that enables *relation-first*, $\mathcal{I}(\vartheta) = \mathcal{I}(\vartheta_2) + \mathcal{I}(\vartheta_1) = 2\mathcal{I}(\vartheta_2) + \mathcal{I}(\vartheta_1 \mid \vartheta_2)$ encapsulates all information needed to "create" the outcome $\mathcal{Y} = \mathcal{Y}_1 + \mathcal{Y}_2$, with $\mathcal{I}(\phi) = 0$ indicating $\phi$ not an informative relation. When adopting the traditional perspective as an observer, $\vartheta_1$ and $\vartheta_2$ simply denote functional parameters, where the *observational information* manifests as $\mathcal{I}(\phi \mid \mathcal{Y}_2) = \mathcal{I}(\mathcal{Y}_1) - \mathcal{I}(\mathcal{Y}_2) > 0$.

For clarity, we use $\vartheta_1 \perp\!\!\!\perp \vartheta_2$ to signify the timing-dimensional independence between $\mathcal{Y}_1$ and $\mathcal{Y}_2$, termed as ***dynamical independence***, without altering the conventional understanding within the observational space, like $Y_1 \perp\!\!\!\perp Y_2 \in \mathbb{R}^m$. On the contrary, $\vartheta_1 \not\!\perp\!\!\!\perp \vartheta_2$ implies a ***dynamical dependence***, which is, an *interaction* between $\mathcal{Y}_1$ and $\mathcal{Y}_2$. "Dynamically dependent or not" only holds when $\mathcal{Y}_1$ and $\mathcal{Y}_2$ are *dynamically significant*.

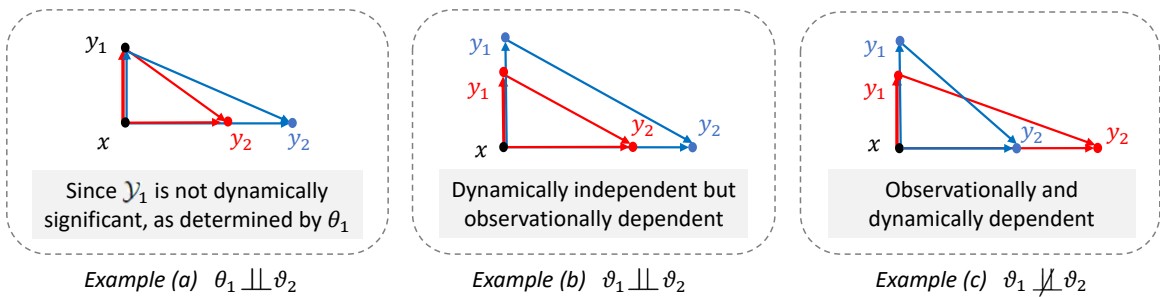

Figure 3: Illustrative examples for dynamical dependence and independence. The observational dependence from $\mathcal{Y}_1$ to $\mathcal{Y}_2$ is displayed as $\overrightarrow{y_1 y_2}$, where red and blue indicate two different data instances.

Figure 3 is upgraded from the conventional causal Directed Acyclic Graph (DAG) in two aspects: 1) A node represents a state value of the variable, and 2) edge length shows timespans for a data instance (i.e., a data point or realization) to achieve this value. This allows for the visualization of dynamic interactions through different data instances. For instance, Figure 3(c) shows that the dependence between $\vartheta_1$ and $\vartheta_2$ inversely impacts their speeds, such that achieving $y_1$ more quickly implies a slower attainment of $y_2$.

## 2.3 Potential Development Toward AGI

As demonstrated, choosing between the observer's or the creator's perspective depends on the questions we are addressing rather than a matter of conflict. In the former, information is gained from observations and

represented by observables; while in the latter, relational information preferentially exists as representing the knowledge we aim to construct in modeling, such that once the model is established, we can use it to deduce outcomes as a description of "possible observations in the future" without direct observation.

Causality questions inherently require the creator's perspective, since "informative observations" cannot emerge out of nowhere. Empirically, it is reflected as the challenge of specifying outcomes in traditional causal modeling, often referred to as "identification difficulty" Zhang (2012). As mentioned by Schölkopf et al. (2021), "we may need a new learning paradigm" to depart from the i.i.d.-based modeling assumption, which essentially asserts the objects we are modeling exactly exist as how we expect them to. We term this conventional paradigm as *object-first* and have introduced the *relation-first* principle accordingly.

| | *No Dynamical Interactions between Learned Outcome Components* | *The Outcome Components present Significant Interactions through $\phi$* |
|---|---|---|
| *Learning Dynamics $do(\cdot)$* | LLMs, Inversed Learning, Reinforcement Learning, Causal Representation Learning | Sequentially perform Relation-First modeling to explore the structuralized dynamic outcome |
| *Only State Outcome $Y$* | | Structural Causal Models, Direct RNN Applications in Causality, Causal Inference, Causal Emergence |

Figure 4: The $do(Y)$-Paradox in traditional Causality Modeling vs. modern Representation Learning.

The *relation-first* thinking has been embraced by the definition of Fisher Information, as well as in do-calculus that differentiates the relational information. Moreover, neural networks with the back-propagation strategy have technologically embodied it. Therefore, it's unsurprising that the advent of AI-based representation learning signifies a turning point in causality modeling. From an engineering standpoint, answering the "what ... if?" (i.e., counterfactual) question indicates the capacity of predicting $do(Y)$ as structuralized dynamic outcomes. Intriguingly, learning dynamics (i.e., the realization of $do(\cdot)$) and predicting outcomes (i.e., facilitating the role of $Y$) present a paradox under the traditional learning paradigm, as in Figure 4.

**About AI-based Dynamical Learning**

Understanding dynamics is a significant instinctive human ability. Representation learning achieves computational optimizations across the timing dimension, notably embodying such capabilities. Specifically, Large Language Models (LLMs) Wes (2023) have sparked discussions about our progress toward AGI Schaeffer et al. (2023). The application of meta-learning Lake & Baroni (2023), in particular, has enabled the autonomous identification of semantically meaningful dynamics, demonstrating the potential for human-like intelligence. Yet, it is also highlighted that LLMs still lack a true comprehension of causality Pavlick (2023).

The complexity of causality lies in potential interactions within a "possible world", not just in computing individual possibilities, whether they are dynamically significant or not. Instead of a single question, "what ... if?" stands for a self-extending logic, where the "if" condition can be applied to computed results repeatedly, leading to complex structures. Thus, causality modeling is to uncover the unobservable knowledge implied by the observable $X/do(X) \rightarrow Y/do(Y)$ phenomenons to enable its outcome beyond direct observations.

Advanced technologies, such as reinforcement learning Arora (2021) and causal representation learning, have blurred the boundary between the roles of variable $X/do(X)$ and outcome $Y/do(Y)$, which are manually maintained in traditional causal inference. They often focus on the advanced efficacy in learning dynamics, yet it is frequently overlooked that the foundational RNN architecture is grounded in $do(X) \rightarrow Y$ without establishing a dynamically interactable $do(Y)$. Essentially, any significant dynamics that are autonomously extracted by AI can be attributed to $do(X)$. Even though within diffusion methods, their computations can be split into multiple rounds of $do(X) \rightarrow Y$, since without an identified meaning as $\mathcal{I}(\vartheta)$, the significance of becoming a $do(Y)$, rather than remaining a sequence of discrete values $Y^\tau = (Y_1, \ldots, Y_\tau)$, is unfounded.

From AI's viewpoint, changes in the values of a sequential variable need not be meaningful, although they may have distinct implications for humans. For instance, a consistent dynamic pattern that varies in unfolding speed might indicate an individual dynamic, $do(X)$, distinct from $X^t$. If this dynamic pattern specifically

signifies the effect (like $\mathcal{I}(\vartheta)$) of a certain cause (like $X/do(X)$), it could represent $do(Y)$. However, if the speed change is attributable to another identifiable effect (such as $\mathcal{I}(\omega)$), it showcases a dynamical interaction.

**About State Outcomes in Causal Inference**

Causal inference and associated Structural Causal Models (SCMs) focus on causal structures, taking into account potential interactions. However, the *object-first* paradigm restricts their outcomes to be "objective observations", represented by $Y_\tau$ with a predetermined timestamp $\tau$. This inherently implies all potential effects conform to a singular "observed timing". Thereby, they can be consolidated into a one-time dynamic, leading to "structuralized observables" instead of "structuralized dynamics". As in Figure 1, the overflowed information $\mathcal{I}(do(Y)) - \mathcal{I}(Y)$ (from an observer's perspective) "emerges" to form an informative relation $\phi$ in a "possible world", rather than a deducible dependence between two dynamics $do(Y)_1$ and $do(Y)_2$.

Such "causal emergence" requires significant efforts on theoretical interpretations. Particularly, the unknown relation $\phi$ is often attributed to the well-known "hidden confounder" problem Greenland et al. (1999); Pearl et al. (2000), linked to the fundamental assumptions of causal sufficiency and faithfulness Sobel (1996). In practice, converting causal knowledge represented by DAGs into operational causal models demands careful consideration Elwert (2013), where data adjustments and model interpretations often rely on human insight Sanchez et al. (2022); Crown (2019). These theoretical accomplishments underpin causal inference's core value in the era dominated by statistical analysis, before the advent of neural networks.

**About Development of Relation-First Paradigm**

As highlighted in Theorem 3, sequential modeling is necessary for causality to achieve structuralized dynamic outcomes. When the prior knowledge of causal structure is given, the relational information $\mathcal{I}(\vartheta)$ has been determined; correspondingly, the sequential input and output data, $x^t = (x_1, \ldots, x_t)$ and $y^\tau = (y_1, \ldots, y_\tau)$, can be chosen to enable AI to extract $\mathcal{I}(\vartheta)$ through them. While for AI-detected meaningful dynamics, we should purposefully recognize "if it suggests a $do(Y)$, what $\mathcal{I}(\vartheta)$ have we extracted?" The gained insights can guide us to make the decision on whether and how to perform the next round of detection based on it.

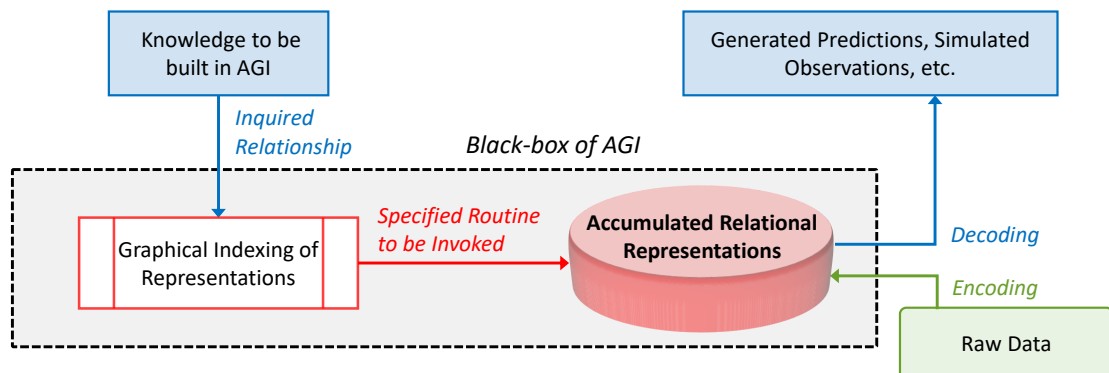

Figure 5: Accessing AGI as a black-box, with human-mediated parts colored in blue. A practically usable system demands long-term representation accumulations and refinements, which mirrors our learning process.

In this way, the relational representations in latent space can be accumulated as vital resources, organized and managed through the graphically structured indices, as depicted in Figure 5. This flow mirrors human learning processes Pitt (2022), with these indices serving as causal DAGs in our comprehension. If knowledge from various domains could be compiled and made accessible like a library over time, then the representation resource might be continuously optimized across diverse scenarios, thereby enhancing generalizability.

From a human standpoint, deciphering latent space representations becomes unnecessary. With sufficient raw data, we have the opportunity to establish nuanced causal reasoning through the use of graphical indices. Specifically, this involves an indexing process that translates inquiries into specific input-output graphical routines, guiding data streaming through autoencoders to produce human-readable observations. Although convenient, this approach could subject computer "intelligence" to more effective control.

# 3   Modeling Framework in Creator's Perspective

Under the traditional i.i.d.-based framework, questions must be addressed individually within their respective modeling processes, even when they share similar underlying knowledge. This necessity arises because each modeling process harbors incorrect premises about the objective reality it faces, which often goes unnoticed because of conventional *object-first* thinking. The advanced modeling flexibility afforded by neural networks further exposes this fundamental issue. Specifically, it is identified as the *model generalizability* challenge by Schölkopf et al. (2021). They introduced the concept of *causal representation* learning, underscoring the importance of prioritizing causal relational information before specifying observables.

Rather than merely raising a new method, we aim to emphasize that the **_shift of perspective_** enables the modeling framework across the "possible timing space" beyond solely observational one. As shown in Figure 6, when adopting the creator's perspective, space $\mathbb{R}^H$ is embraced to accommodate the abstract variables representing the informative relations, where the notion of $\omega$ will be introduced later.

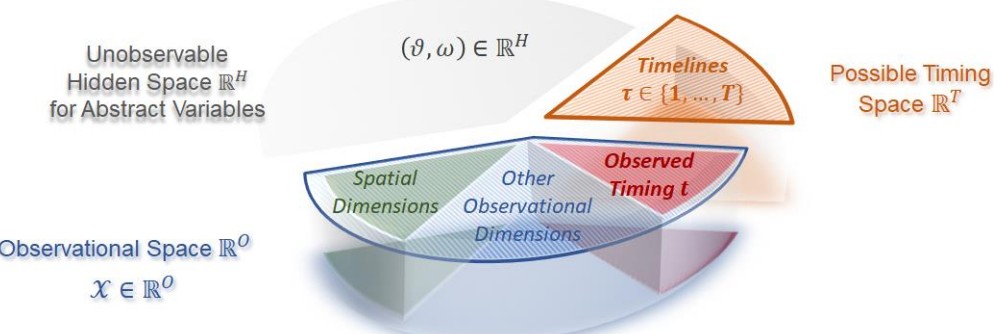

Figure 6: The framework from the creator's perspective, where $\mathcal{Y} \in \mathbb{R}^{O-1} \cup \mathbb{R}^T$ (with $t$ excluded) represents the outcome governed by $\mathcal{I}(\vartheta, \omega)$, without implying any observational information. An observer's perspective is $\mathcal{Y} \in \mathbb{R}^{O-1} \cup \tau$, with the observational information $\mathcal{I}(\mathcal{Y})$ defined, but without $\mathbb{R}^H$ or $\mathbb{R}^T$ perceived.

When adopting an observer's perspective, it involves answering a "what...if" question just once. However, the genesis of such questions is rooted in the perspective of a "creator", aiming to explore all possibilities for the optimal choice, which is precisely what we embrace when seeking technological or engineering solutions.

Every possibility represents an observational outcome ("the what...") for a specific causal relationship ("the if...") or a routine of consecutive relationships within a DAG, akin to placing an observer within the creator's conceptual space. Thus, the "creator's perspective" acts as a space encompassing all potential "observer's perspectives" by treating the latter as a variable. Within this framework, the once perplexing concept of "collapse" in quantum mechanics becomes readily understandable.

From the creator's perspective, a causal relationship $\mathcal{X} \xrightarrow{\vartheta} \mathcal{Y}$ suggests that $\mathcal{Y}$ belongs to $\mathbb{R}^{O-1} \cup \mathbb{R}^T$, where $\mathbb{R}^T$ represents a $T$-dimensional space with timing $\tau = 1, \ldots, T$ sequentially marking the $T$ components of $\mathcal{Y}$. The separation of these components depends on the creator's needs, regardless of which, their aggregate, $\mathcal{Y} = \sum_{\tau=1}^{T} \mathcal{Y}_\tau$, is invariably governed by $\mathcal{I}(\vartheta)$. However, once the creator places an observer for this relationship, from this "newborn" observer's viewpoint, space $\mathbb{R}^T$ ceases to exist and is perceived solely as an "observed timeline" $\tau$. In other words, $\tau$ has lost its computational flexibility as the "timing dimension" but remains merely a sequence of constant timestamps.

Thus, the term "collapse" refers to this singular "perspective shift". Metaphorically, a one-time "collapse" is akin to opening Schrödinger's box once, and in the modeling context, it signifies that a singular modeling computation has occurred. Accordingly, Theorem 3 can be reinterpreted: Causality modeling is to facilitate "structuralized collapses" within $\mathbb{R}^T$ from the creator's perspective. Importantly, for the creator, $\mathbb{R}^T$ is not limited to representing a single relationship but can also include "structuralized relationships" by embracing a broader macro-level perspective. In light of this, we introduce the following definitions.

**Definition 4.** A causal relation $\vartheta$ can be defined as **_micro-causal_** if an extraneous relation $\omega$ exists, where $\mathcal{I}(\omega) \not\subseteq \mathcal{I}(\vartheta)$, such that incorporating $\omega$ can form a new, **_macro-causal_** relation, denoted by $(\vartheta, \omega)$. The process of incorporating $\omega$ is referred to as a **_generalization_**.

**Definition 5.** From the creator's perspective, the **_macro-level_** possible timing space $\mathbb{R}^T = \sum_{\tau=1}^{T} \mathbb{R}^\tau$ is constructed by aggregating each **_micro-level_** space $\mathbb{R}^\tau$, where $\tau \in \{1, \ldots, T\}$ indicates the **_timeline_** that houses the sequential timestamps by adopting the observer's perspective for $\mathbb{R}^\tau$.

To clarify, the $T$-dimensional space $\mathbb{R}^T$ mentioned earlier is considered a micro-level concept, which we formally denote as $\mathbb{R}^\tau$. Upon transitioning to the macro-level possible timing space $\mathbb{R}^T$, the creator's perspective is invoked. Within this perspective, both $\mathbb{R}^H$ and $\mathbb{R}^T$ are viewed as conceptual spaces, lacking computationally meaningful notions like "dimensionality" or specific "distributions".

In essence, the moment we contemplate a potential "computation", the observer's perspective is already established, from which, the micro-level space $\mathbb{R}^\tau$ (or a collection of such spaces $\{\mathbb{R}^\tau\}$) has been defined and "primed for collapse" through the methodologies under contemplation. Philosophically, the notion of a timeline $\tau$ within the "thought space" $\mathbb{R}^T$ is characterized as "relative timing" Wulf et al. (1994); Shea et al. (2001), in contrast to the "absolute timing" represented by $t$ in this paper. Moreover, in the modeling context, computations involving $\tau$ can draw upon the established Granger causality approach Granger (1993).

### 3.1 Hierarchical Levels by $\omega$

As illustrated in Figure 1, the "causal emergence" phenomenon stems from adopting different perspectives, not truly integrating new relational information. We employ the terms "micro-causal" and "macro-causal" to identify the new information integration, defining the *generalization* process (as per Definition 4), and its inverse is termed *individualization*. In modeling, the **_generalizability_** of an established micro-causal model $f(; \vartheta)$ is its ability to be reused in macro-causality without diminishing $\mathcal{I}(\vartheta)$'s representation.

The information gained from $\mathcal{I}(\vartheta)$ to $\mathcal{I}(\vartheta, \omega)$ often introduces a new **_hierarchical level_** of relation, thereby raising generalizability requirements for causal models. This may suggest new observables, potentially as new causes or outcome components, or both. Let's consider a *logically* causal relationship (without such significance in modeling) as a simple example: Family incomes $X$ affecting grocery shopping frequencies $Y$, represented as $X \xrightarrow{\theta} Y$, where $\theta$ may vary internationally due to cultural differences $\omega$, creating two levels: a global-level $\theta$ and a country-level $(\theta \mid \omega)$. While $\omega$ isn't a direct modeling target, it's an essential condition, necessitating the total information $\mathcal{I}(\theta, \omega) = \mathcal{I}(\theta \mid \omega) + \mathcal{I}(\omega)$. From the observer's perspective, it equates to incorporating an additional observable, like country $Z$, as a new cause to affect $Y$ with $X$ jointly.

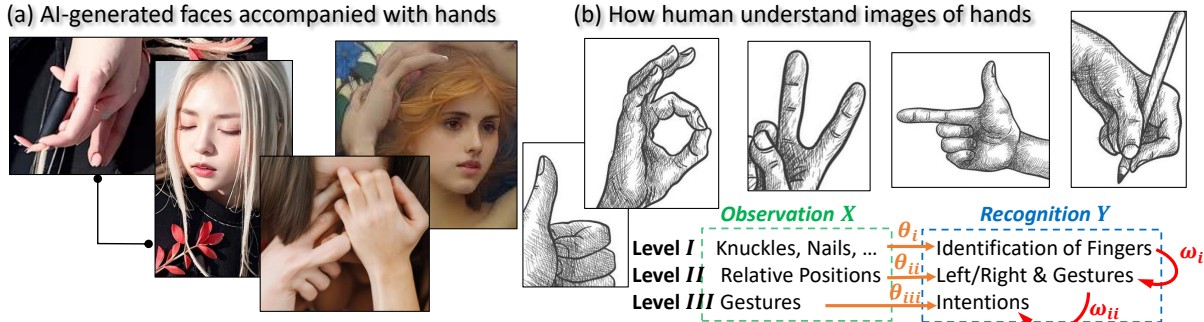

Figure 7: AI can generate reasonable faces but treat hands as arbitrary mixtures of fingers, while humans understand observations hierarchically to avoid mess, sequentially indexing through $\{\theta_i, \theta_{ii}, \theta_{iii}\}$.

Addressing hierarchies within knowledge is a common issue in relationship modeling, but timing distributional hierarchies present significant challenges to traditional methods, leading to the development of a specialized "group-specific learning" Fuller et al. (2007), which primarily depends on manual identifications. However,

this approach is no longer viable in modern AI-based applications, necessitating the adoption of the *relation-first* modeling paradigm. Below, we present two examples to demonstrate this necessity: one is solely observational, and the other involves a causality with timing hierarchy.

**Observational Hierarchy Example**

The AI-created personas on social media can have realistic faces but seldom showcase hands, since AI struggles with the intricate structure of hands, instead treating them as arbitrary assortments of finger-like items. Figure 7(a) shows AI-created hands with faithful color but unrealistic shapes, while humans can effortlessly discern hand gestures from the grayscale sketches in (b).

Human cognition intuitively employs informative relations as the *indices* to visit mental representations Pitt (2022). As in (b), this process operates hierarchically, where each higher-level understanding builds upon conclusions drawn at preceding levels. Specifically, Level **I** identifies individual fingers; Level **II** distinguishes gestures based on the positions of the identified fingers, incorporating additional information from our understanding of how fingers are arranged to constitute a hand, denoted by $\omega_i$; and Level **III** grasps the meanings of these gestures from memory, given additional information $\omega_{ii}$ from knowledge.

Conversely, AI models often do not distinguish the levels of relational information, instead modeling overall as in a relationship $X \xrightarrow{\theta} Y$ with $\theta = (\theta_i, \theta_{ii}, \theta_{iii})$, resulting a lack of informative insights into $\omega$. However, the hidden information $\mathcal{I}(\omega)$ may not always be essential. For example, AI can generate convincing faces because the appearance of eyes $\theta_i$ strongly indicates the facial angles $\theta_{ii}$, i.e., $\mathcal{I}(\theta_{ii}) = \mathcal{I}(\theta_i)$ indicating $\mathcal{I}(\omega_i) = 0$, removing the need to distinguish eyes from faces.

On the other hand, given that $X$ has been fully observed, AI can inversely deduce the relational information using methods such as reinforcement learning Sutton (2018); Arora (2021). In this particular case, when AI receives approval for generating hands with five fingers, it may autonomously begin to derive $\mathcal{I}(\theta_i)$. However, when such hierarchies occur on the timing dimension of a dynamically significant $\mathcal{Y}$, they can hardly be autonomously identified, regardless of whether AI techniques are leveraged.

**Timing Hierarchy in Causality Example**

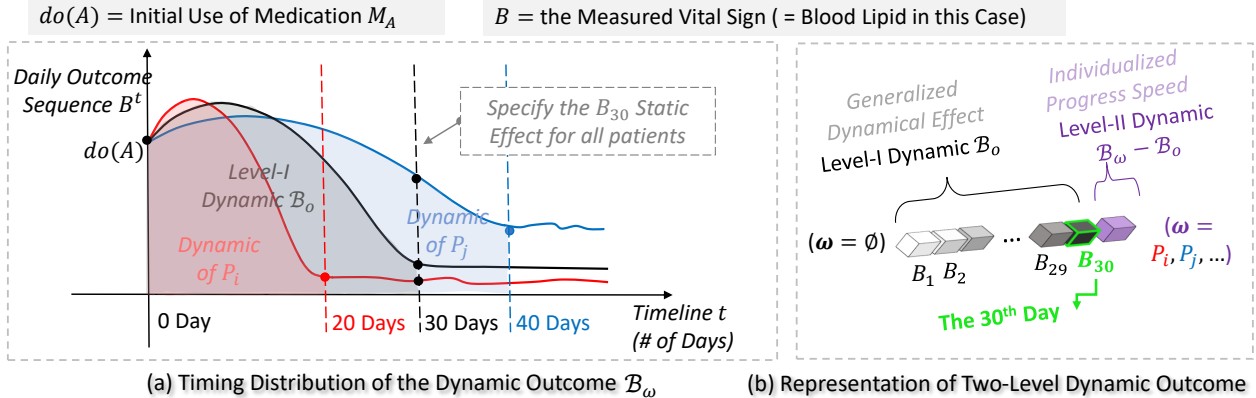

(a) Timing Distribution of the Dynamic Outcome $\mathcal{B}_\omega$      (b) Representation of Two-Level Dynamic Outcome

Figure 8: $do(A) =$ the initial use of medication $M_A$ for reducing blood lipid $B$. By the rule of thumb, the effect of $M_A$ needs around 30 days to fully release ($t = 30$ at the black curve elbow). Patient $P_i$ and $P_j$ achieve the same magnitude of the effect by 20 and 40 days instead.

In Figure 8, $\mathcal{B}_\omega$ represents the observational sequence $B^t = (B_1, \ldots, B_{30})$ from a group of patients identified by $\omega$. Clinical studies typically aim to estimate the average effect (generalized-level I) on a predetermined day, like $B_{t+30} = f(do(A_t))$. However, our inquiry is indeed the complete level I dynamic $\mathcal{B}_o = \int_{t=1}^{30} do(B_t)dt$, which describes the trend of effect changing over time, without anchored timestamps. To eliminate the level II dynamic from data, a "hidden confounder" is usually introduced to represent their unobserved personal characteristics. Let us denote it by $E$, and assume $E$ linearly impact $\mathcal{B}_o$, making the level II dynamic $\mathcal{B}_\omega - \mathcal{B}_o$ simply signifying their individualized progress speeds for the same effect $\mathcal{B}_o$.

To accurately represent $\mathcal{B}_o$ with a sequential outcome, traditional methods necessitate an intentional selection or adjustment of training data. This is to ensure the "influence of $E$" is eliminated from the data, even unavoidable when adopting RNN models. In RNNs, the dynamically significant representation is facilitated only on $do(A)$, while the sequential outcome $B^t$ still requires predetermined timestamps. However, once $t$ is specified for all patients without the data selection - for example, let $t = 30$ to snapshot $B_{30}$ - bias is inherently introduced, since $B_{30}$ represents the different magnitude of effect $\mathcal{B}_o$ for various patients.

Such hierarchical dynamic outcomes are prevalent in many fields, such as epidemic progression, economic fluctuations, and strategic decision-making. Causal inference typically requires intentional data preprocessing to mitigate inherent biases, including approaches like PSM Benedetto et al. (2018) and backdoor adjustment Pearl (2009), essentially to identify the targeted levels manually. However, they have become impractical due to the modern data volume, and also pose a risk of significant information loss snowballing in structuralized relationship modeling. On the other hand, the significance of timing hierarchies has prompted the development of neural network-based solutions in fields like anomaly detection Wu et al. (2018) to address specific concerns without the intention of establishing a causal modeling framework.

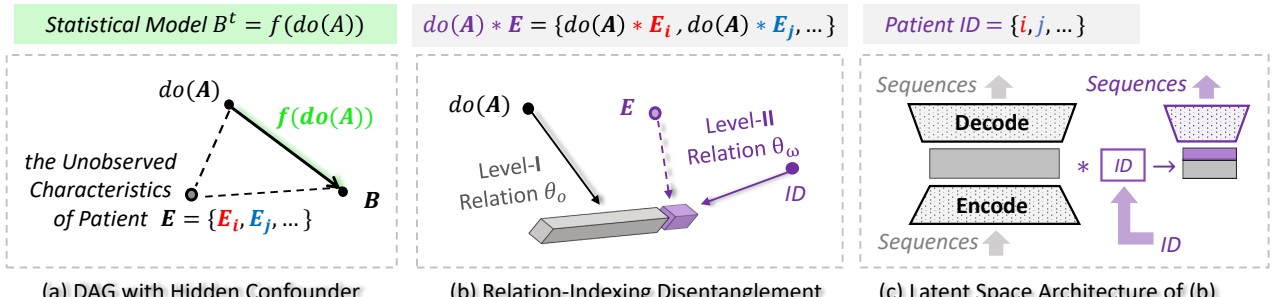

(a) DAG with Hidden Confounder  (b) Relation-Indexing Disentanglement  (c) Latent Space Architecture of (b)

Figure 9: (a) shows the traditional causal DAG for the scenario depicted in Figure 8, (b) disentangles its dynamic outcome in a hierarchical way by indexing through relations, and (c) briefly illustrates the autoencoder architecture for realizing the generalized and individualized reconstructions, respectively.

The concept of "hidden confounder" is essentially elusive, acting more as an interpretational compensation rather than a constructive effort to enhance the model. For example, Figure 9 (a) shows the conventional causal DAG with hidden $E$ depicted. Although the "personal characteristics" are signified, it is not required to be revealed by collecting additional data. This leads to an illogical implication: "Our model is biased due to some unknown factors we don't intend to know." Indeed, this strategy employs a hidden observable to account for the omitted timing-dimensional nonlinearities in statistical models.

As illustrated in Figure 9(b), the associative causal variable $do(A) * E$ remains unknown, unable to form a modelable relationship. On the other hand, *relation-first* modeling approaches only require an observed identifier to index the targeted level in representation extractions, like the patient ID denoted by $\omega$.

## 3.2   The Generalizability Challenge across Multiple Timelines in $\mathbb{R}^T$

From the creator's perspective, timelines in the macro-level possible timing space $\mathbb{R}^T$ may pertain to different micro-causalities, implying "structuralized" causal relationships. This poses a significant generalizability challenge for traditional structural causal models (SCMs).

The example in Figure 10 showcases a practical scenario in a clinical study. This 3D causal DAG includes two timelines, $\tau_\theta$ and $\tau_\omega$, with the $x$-axis categorically arranging observables. The upgrades to causal DAGs, as applied in Figure 3, are also adopted here, ensuring that the lengths of the arrows reflect the timespan required to achieve the state values represented by the observable nodes. Here, the nodes marked in uppercase letters indicate the values representing the mean effects of the current data population, i.e., the group of patients under analysis. Accordingly, the lengths of the arrows indicate their mean timespans.

We use $\Delta\tau_\theta$ and $\Delta\tau_\omega$ to signify the time steps (i.e., the unit timespans) on $\tau_\theta$ and $\tau_\omega$, respectively. Considering the triangle $SA'B'$, when each unit of effect is delivered from $S$ to $A'$ (taking $\Delta\tau_\omega$), it immediately starts

impacting $B'$ through $\overrightarrow{A'B'}$ (with $\Delta\tau_\theta$ required); simultaneously, the next unit of effect begins its generation at $S$. Under the *relation-first* principle, this dual action requires a two-step modeling process to sequentially extract the dynamic representations on $\tau_\theta$ and $\tau_\omega$. However, in traditional SCM, it is represented by the edge $\overrightarrow{SB'}$ with a priorly specified timespan from $S$ to $B'$. This inherently sets the $\Delta\tau_\theta : \Delta\tau_\omega$ ratio based on the current population's performance, freezing the state value represented by $B'$ and fixing the geometrical shape of the $ASB'$ triangle in this space.

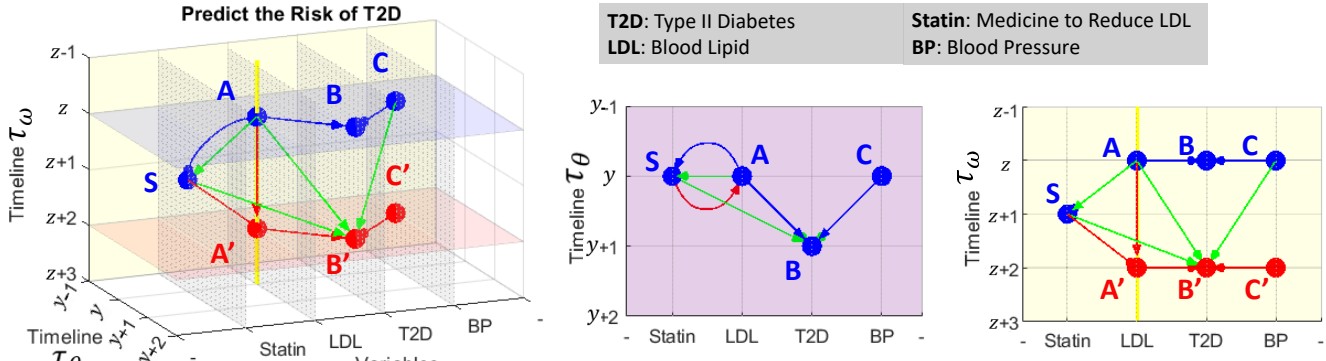

Figure 10: A 3D-view DAG in $\mathbb{R}^{O-1} \cup \mathbb{R}^T$ with two timelines $\tau_\theta$ and $\tau_\omega$. The SCM $\boxed{B' = f(A, C, S)}$ is to evaluate the effect of Statin on reducing T2D risks. On $\tau_\theta$, the step $\Delta\tau_\theta$ from $y$ to $(y+1)$ allows $A$ and $C$ to fully influence $B$; the step $\Delta\tau_\omega$ on $\tau_\omega$ from $(z+1)$ to $(z+2)$ let $S$ fully release to forward status $A$ to $A'$.

The lack of model generalizability manifests in various ways, depending on the intended scale of generalization. For instance, when focusing on a finer micro-scale causality, the SCM that describes the mean effects for the current population cannot be tailored to individual patients within this population. Conversely, aiming to generalize this SCM to accommodate other populations, or a broader macro-scale causality, may lead to failure because the preset $\Delta\tau_\theta : \Delta\tau_\omega$ ratio lacks universal applicability.

### 3.3 Fundamental Reliance on Assumptions under Object-First

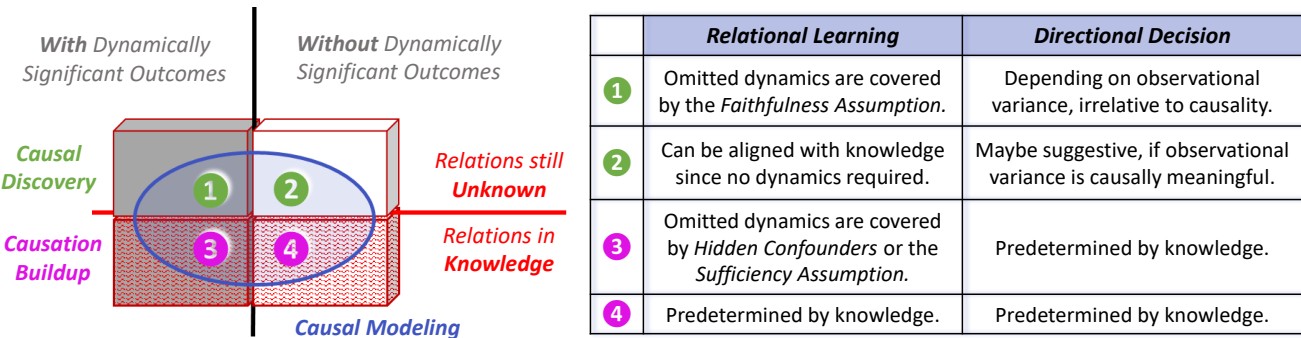

| | | Relational Learning | Directional Decision |
|---|---|---|---|
| 1 | | Omitted dynamics are covered by the *Faithfulness Assumption*. | Depending on observational variance, irrelative to causality. |
| 2 | | Can be aligned with knowledge since no dynamics required. | Maybe suggestive, if observational variance is causally meaningful. |
| 3 | | Omitted dynamics are covered by *Hidden Confounders* or the *Sufficiency Assumption*. | Predetermined by knowledge. |
| 4 | | Predetermined by knowledge. | Predetermined by knowledge. |

Figure 11: Categories of causal modeling applications. The left rectangular cube indicates all *logically causal* relationships, with the blue circle indicating potentially modelable ones.

Figure 11 categorizes the current causal model applications based on two aspects: 1) if the structure of $\theta/\vartheta$ is known a priori, they are used for structural causation buildup or causal discovery; 2) depending on whether the required outcome is dynamically significant, they can either accurately represent true causality or not.

Under the conventional modeling paradigm, capturing the significant dynamics within causal outcomes autonomously is challenging. When building causal models based on given prior knowledge, the omitted dynamics become readily apparent. If these dynamics can be specifically attributed to certain unobserved

observables, like the node $E$ in Figure 9(a), such information loss is attributed to a hidden confounder. Otherwise, they might be overlooked due to the *causal sufficiency* assumption, which presumes that all potential confounders have been observed within the system. Typical examples of approaches susceptible to these issues are structural equation models (SEMs) and functional causal models (FCMs) Glymour et al. (2019); Elwert (2013). Although state-of-the-art deep learning applications have effectively transformed the discrete structural constraint into continuous optimizations Zheng et al. (2018; 2020); Lachapelle et al. (2019), issues of lack of generalizability still hold Schölkopf et al. (2021); Luo et al. (2020); Ma et al. (2018).

On the other hand, causal discovery primarily operates within the $\mathbb{R}^O$ space and is incapable of detecting dynamically significant causal outcomes. If the interconnection of observables can be accurately specified as the functional parameter $\theta$, there remains a chance to discover informative correlations. Otherwise, mere conditional dependencies among observables are unreliable for causal reasoning, as seen in Bayesian networks Pearl et al. (2000); Peters et al. (2014). Typically, undetected dynamics are overlooked due to the *Causal Faithfulness* assumption, which suggests that the observables can fully represent the underlying causal reality.

Furthermore, the causal directions suggested by the results of causal discovery often lack logical causal implications. Consider $X$ and $Y$ in the optional models $Y = f(X; \theta)$ and $X = g(Y; \phi)$, with predetermined parameters, which indicate opposite directions. Typically, the direction $X \to Y$ would be favored if $\mathcal{L}(\hat{\theta}) > \mathcal{L}(\hat{\phi})$. Let $\mathcal{I}_{X,Y}(\theta)$ denote the information about $\theta$ given $\mathbf{P}(X, Y)$. Using $p(\cdot)$ as the density function, the integral $\int_X p(x; \theta) dx$ remains constant in this context. Then:

$$\mathcal{I}_{X,Y}(\theta) = \mathbb{E}[(\frac{\partial}{\partial \theta} \log p(X, Y; \theta))^2 \mid \theta] = \int_Y \int_X (\frac{\partial}{\partial \theta} \log p(x, y; \theta))^2 p(x, y; \theta) dx dy$$

$$= \alpha \int_Y (\frac{\partial}{\partial \theta} \log p(y; x, \theta))^2 p(y; x, \theta) dy + \beta = \alpha \mathcal{I}_{Y|X}(\theta) + \beta, \text{with } \alpha, \beta \text{ being constants.}$$

$$\text{Then, } \hat{\theta} = \arg\max_\theta \mathbf{P}(Y \mid X, \theta) = \arg\min_\theta \mathcal{I}_{Y|X}(\theta) = \arg\min_\theta \mathcal{I}_{X,Y}(\theta), \text{ and } \mathcal{L}(\hat{\theta}) \propto 1/\mathcal{I}_{X,Y}(\hat{\theta}).$$

The inferred directionality indicates how informatively the observational data distribution can reflect the two predetermined parameters. Consequently, such directionality is unnecessarily logically meaningful but could be dominated by the data collection process, with the predominant entity deemed the "cause", consistent with other existing conclusions Reisach et al. (2021); Kaiser & Sipos (2021).

## 4 Relation-Indexed Representation Learning (RIRL)

This section introduces a method for realizing the proposed *relation-first* paradigm, referred to as RIRL for brevity. Unlike existing causal representation learning, which is primarily confined to the micro-causal scale, RIRL focuses on facilitating *structural causal dynamics exploration* in the latent space.

Specifically, "relation-indexed" refers to its micro-causal realization approach, guided by the *relation-first* principle, where the indexed representations are capable of capturing the dynamic features of causal outcomes across their timing-dimensional distributions. Furthermore, from a macro-causal viewpoint, the extracted representations naturally possess high generalizability, ready to be reused and adapted to various practical conditions. This advancement is evident in the structural exploration process within the latent space.

Unlike traditional causal discovery, RIRL exploration spans $\mathbb{R}^{O-1} \cup \mathbb{R}^T$ to detect causally significant dynamics without concerns about "hidden confounders", where $\mathbb{R}^T$ encompasses all possibilities of the potential causal structure. The representations obtained in each round of RIRL detection serve as elementary units for reuse, enhancing the flexibility of structural models. This exploration process eventually yields DAG-structured graphical indices, with each input-output pair representing a specific causal routine, readily accessible.

Subsequently, section 4.1 delves into the micro-causal realization to discuss the technical challenges and their resolutions, including the architecture and core layer designs. Section 4.2 introduces the process of "stacking" relation-indexed representations in the latent space, to achieve hierarchical disentanglement at an effect node in DAG. Finally, section 4.3 demonstrates the exploration algorithm from a macro-causal viewpoint.

### 4.1 Micro-Causal Architecture

For a relationship $\mathcal{X} \xrightarrow{\theta} \mathcal{Y}$ given sequential observations $\{x^t\}$ and $\{y^\tau\}$, with $|\overrightarrow{x}| = n$ and $|\overrightarrow{y}| = m$, the relation-indexed representation aims to establish $(\mathcal{X}, \theta, \mathcal{Y})$ in the latent space $\mathbb{R}^L$. Firstly, an *initialization* is needed for $\mathcal{X}$ and $\mathcal{Y}$ individually, to construct their latent space representations from observed data sequences. For clarity, we use $\mathcal{H} \in \mathbb{R}^L$ and $\mathcal{V} \in \mathbb{R}^L$ to refer to the latent representations of $\mathcal{X} \in \mathbb{R}^O$ and $\mathcal{Y} \in \mathbb{R}^O$, respectively. The neural network optimization to derive $\theta$ is a procedure between $\mathcal{H}$ as input and $\mathcal{V}$ as output. In each iteration, $\mathcal{H}$, $\theta$, and $\mathcal{V}$ are sequentially refined in three steps, until the distance between $\mathcal{H}$ and $\mathcal{V}$ is minimized within $\mathbb{R}^L$, without losing their representations for $\mathcal{X}$ and $\mathcal{Y}$. Consider instances $x$ and $y$ of $\mathcal{X}$ and $\mathcal{Y}$ that are represented by $h$ and $v$ correspondingly in $\mathbb{R}^L$, as in Figure 14. The latent dependency $\mathbf{P}(v|h)$ represents the relational function $f(;\theta)$. The three optimization steps are as follows:

1. Optimizing the cause-encoder by $\mathbf{P}(h|x)$, the relation model by $\mathbf{P}(v|h)$, and the effect-decoder by $\mathbf{P}(y|v)$ to reconstruct the relationship $x \to y$, represented as $h \to v$ in $\mathbb{R}^L$.
2. Fine-tuning the effect-encoder $\mathbf{P}(v|y)$ and effect-decoder $\mathbf{P}(y|v)$ to accurately represent $y$.
3. Fine-tuning the cause-encoder $\mathbf{P}(h|x)$ and cause-decoder $\mathbf{P}(x|h)$ to accurately represent $x$.

In this process, $h$ and $v$ are iteratively adjusted to reduce their distance in $\mathbb{R}^L$, with $\theta$ serving as a bridge to span this distance and guiding the output to fulfill the associated representation $(\mathcal{H}, \theta, \mathcal{V})$. From the perspective of the effect node $\mathcal{Y}$, this tuple represents its component indexing through $\theta$, denoted as $\mathcal{Y}_\theta$.

However, it introduces a technical challenge: for a micro-causality $\theta$, the dimensionality $L$ of the latent space must satisfy $L \geq rank(\mathcal{X}, \theta, \mathcal{Y})$ to provide adequate freedom for computations. To accommodate a structural DAG, this lower boundary can be further enhanced, to be certainly larger than the input vector length $|\overrightarrow{\mathcal{X}}| = t * n$. This necessitates a specialized autoencoder to realize a "higher-dimensional representation", where the accuracy of its reconstruction process becomes significant, and essentially requires *invertibility*.

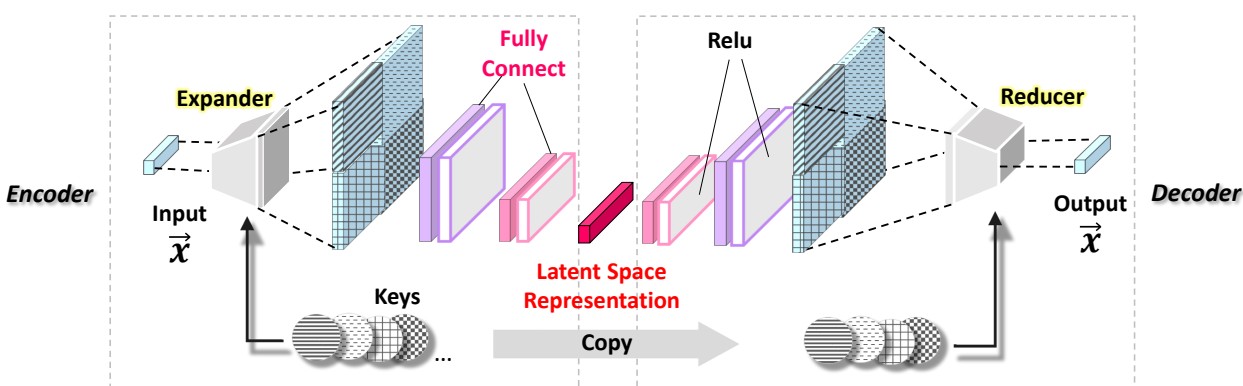

Figure 12: *Invertible* autoencoder architecture for extracting *higher-dimensional* representations.

Figure 12 illustrates the designed autoencoder architecture, featured by a pair of symmetrical layers, named *Expander* and *Reducer* (source code is available [1]). The Expander magnifies the input vector by capturing its higher-order associative features, while the Reducer symmetrically diminishes dimensionality and reverts to its initial formation. For example, the Expander showcased in Figure 12 implements a *double-wise* expansion. Every duo of digits from $\overrightarrow{\mathcal{X}}$ is encoded into a new digit by associating with a random constant, termed the *Key*. This *Key* is generated by the encoder and replicated by the decoder. Such pairwise processing of $\overrightarrow{\mathcal{X}}$ expands its length from $(t * n)$ to be $(t * n - 1)^2$. By concatenating the expanded vectors using multiple *Keys*, $\overrightarrow{\mathcal{X}}$ can be considerably expanded, ready for the subsequent reduction through a regular encoder.

The four blue squares in Figure 12 with unique grid patterns signify the resultant vectors of the four distinct *Keys*, with each square symbolizing a $(t * n - 1)^2$ length vector. Similarly, higher-order expansions, such as *triple-wise* across three digits, can be chosen with adapted *Keys* to achieve more precise reconstructions.

---

[1]https://github.com/kflijia/bijective_crossing_functions/blob/main/code_bicross_extracter.py

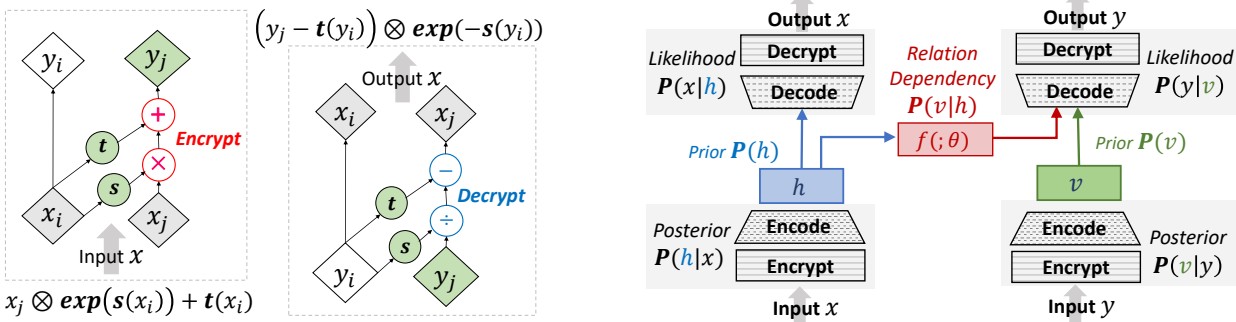

Figure 13: Expander (left) and Reducer (right).

Figure 14: Micro-Causal architecture.

Figure 13 illustrates the encoding and decoding processes within the Expander and Reducer, targeting the digit pair $(x_i, x_j)$ for $i \neq j \in 1, \ldots, n$. The Expander function is defined as $\eta_\kappa(x_i, x_j) = x_j \otimes exp(s(x_i)) + t(x_i)$, which hinges on two elementary functions, $s(\cdot)$ and $t(\cdot)$. The parameter $\kappa$ represents the adopted *Key* comprising of their weights $\kappa = (w_s, w_t)$. Specifically, the Expander morphs $x_j$ into a new digit $y_j$ utilizing $x_i$ as a chosen attribute. In contrast, the Reducer symmetrically performs the inverse function $\eta_\kappa^{-1}$, defined as $(y_j - t(y_i)) \otimes exp(-s(y_i))$. This approach circumvents the need to compute $s^{-1}$ or $t^{-1}$, thereby allowing more flexibility for nonlinear transformations through $s(\cdot)$ and $t(\cdot)$. This is inspired by the groundbreaking work in Dinh et al. (2016) on invertible neural network layers employing bijective functions.

## 4.2   Stacking Relation-Indexed Representations

In each round of detection during the macro-causal exploration, a micro-causal relationship will be selected for establishment. Nonetheless, the cause node in it may have been the effect node in preceding relations, e.g., the component $\mathcal{Y}_\theta$ may already exist at $\mathcal{Y}$ when $\mathcal{Y} \to \mathcal{Z}$ is going to be established. This process of conditional representation buildup is referred to as "stacking".

For a specific node $\mathcal{X}$, the stacking processes, where it serves as the effect, sequentially construct its hierarchical disentanglement according to the DAG. It requires the latent space dimensionality to be larger than $rank(X) + T$, where $T$ represents the in-degree of node $\mathcal{X}$ in this DAG, as well as its number of components as the dynamic effects. From a macro-causal perspective, $T$ can be viewed as the number of necessary edges in a DAG. While to fit it into $\mathbb{R}^L$, a predetermined $L$ must satisfy $L > rank(\mathbf{X}) + T$, where $\mathbf{X}$ represents the data matrix encompassing all observables. In this study, we bypass further discussions on dimensionality boundaries by assuming $L$ is large enough for exploration, and empirically determine $L$ for the experiments.

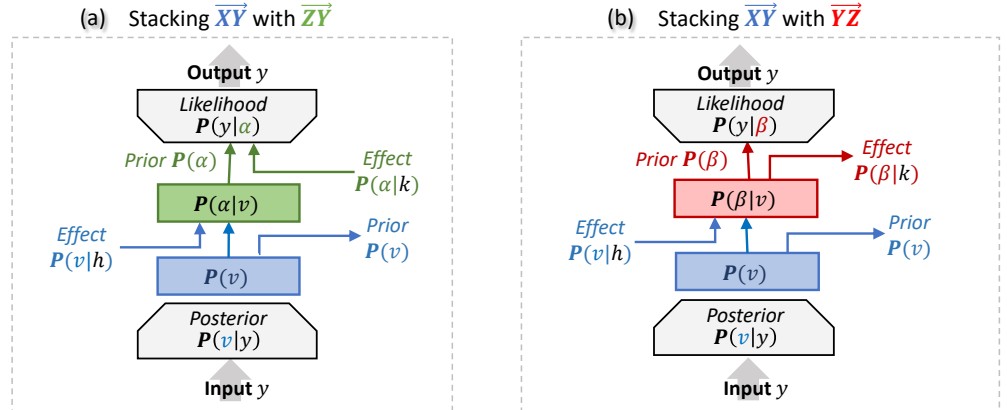

Figure 15: Stacking relation-indexed representations to achieve hierarchical disentanglement.

Figure 15 illustrates the stacking architectures under two different scenarios within a three-node system $\{\mathcal{X}, \mathcal{Y}, \mathcal{Z}\}$. In this figure, the established relationship $\mathcal{X} \to \mathcal{Y}$ is represented by the blue data streams and

layers. The scenarios differ in the causal directions between $\mathcal{Y}$ and $\mathcal{Z}$: the left side represents $\mathcal{X} \to \mathcal{Y} \leftarrow \mathcal{Z}$, while the right side depicts $\mathcal{X} \to \mathcal{Y} \to \mathcal{Z}$.

The hierarchically stacked representations allow for flexible input-output combinations to represent different causal routines as needed. For simple exemplification, we use $\mapsto$ to denote the input and output layers in the stacking architecture. On the left side of Figure 15, $\mathbf{P}(v|h) \mapsto \mathbf{P}(\alpha)$ represents the $\mathcal{X} \to \mathcal{Y}$ relationship, while $\mathbf{P}(\alpha|k)$ implies $\mathcal{Z} \to \mathcal{Y}$. Conversely, on the right, $\mathbf{P}(v) \mapsto \mathbf{P}(\beta|k)$ denotes the $\mathcal{Y} \to \mathcal{Z}$ relationship with $\mathcal{Y}$ as the input. Meanwhile, $\mathbf{P}(v|h) \mapsto \mathbf{P}(\beta|k)$ captures the causal sequence $\mathcal{X} \to \mathcal{Y} \to \mathcal{Z}$.

### 4.3 Exploration Algorithm in the Latent Space

---
**Algorithm 1:** RIRL Exploration

---
**Result:** ordered edges set $\mathbf{E} = \{e_1, \ldots, e_n\}$
$\mathbf{E} = \{\}$ ; $N_R = \{n_0 \mid n_0 \in N, Parent(n_0) = \varnothing\}$ ;
**while** $N_R \subset N$ **do**

  $\Delta = \{\}$ ;
  **for** $n \in N$ **do**
    **for** $p \in Parent(n)$ **do**
      **if** $n \notin N_R$ $and$ $p \in N_R$ **then**
        $e = (p, n)$;
        $\beta = \{\}$;
        **for** $r \in N_R$ **do**
          **if** $r \in Parent(n)$ $and$ $r \neq p$ **then**
            $\beta = \beta \cup r$
          **end**
        **end**
        $\delta_e = K(\beta \cup p, n) - K(\beta, n)$;
        $\Delta = \Delta \cup \delta_e$;
      **end**
    **end**
  **end**
  $\sigma = argmin_e(\delta_e \mid \delta_e \in \Delta)$;
  $\mathbf{E} = \mathbf{E} \cup \sigma$;   $N_R = N_R \cup n_\sigma$;
**end**

---

| | |
|---|---|
| $G = (N, E)$ | graph $G$ consists of $N$ and $E$ |
| $N$ | the set of nodes |
| $E$ | the set of edges |
| $N_R$ | the set of reachable nodes |
| $\mathbf{E}$ | the list of discovered edges |
| $K(\beta, n)$ | KLD metric of effect $\beta \to n$ |
| $\beta$ | the cause nodes |
| $n$ | the effect node |
| $\delta_e$ | KLD Gain of candidate edge $e$ |
| $\Delta = \{\delta_e\}$ | the set $\{\delta_e\}$ for $e$ |
| $n, p, r$ | notations of nodes |
| $e, \sigma$ | notations of edges |

Algorithm 1 outlines the heuristic exploration procedure among the initialized representations of nodes. We employ the Kullback-Leibler Divergence (KLD) as the optimization criterion to evaluate the similarity between outputs, such as the relational $\mathbf{P}(v|h)$ and the prior $\mathbf{P}(v)$. A lower KLD value indicates a stronger causal strength between the two nodes. Additionally, we adopt the Mean Squared Error (MSE) as another measure of accuracy. Considering its sensitivity to data variances Reisach et al. (2021), we do not choose MSE as the primary criterion.

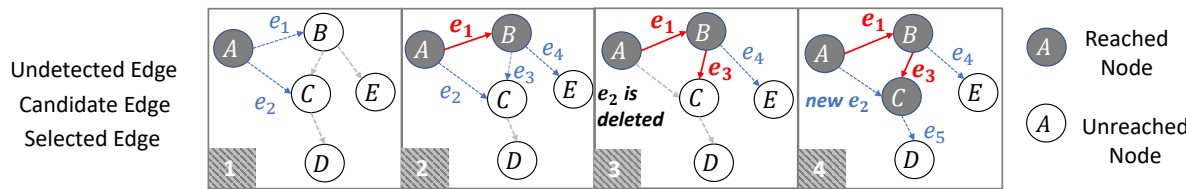

Figure 16: An illustrative example of a detection round in latent space during RIRL exploration.

Figure 16 completely illustrates a detection round within the latent space that represents $\mathbb{R}^{O-1} \cup \mathbb{R}^T$. A new representation for the selected edge is stacked upon the previously explored causal structure during this process. It contains four primary steps: In Step 1, two edges, $e_1$ and $e_3$, have been selected in previous detection rounds. In Step 2, $e_1$, having been selected, becomes the preceding effect at node $B$ for the next round. In Step 3, with $e_3$ selected in the new round, the candidate edge $e_2$ from $A$ to $C$ must be deleted and rebuilt since $e_3$ alters the conditions at $C$. Step 4 depicts the resultant structure.

## 5 RIRL Exploration Experiments

In the experiments, our objective is to evaluate the proposed RIRL method from three perspectives: 1) the performance of the higher-dimensional representation autoencoder, assessed through its reconstruction accuracy; 2) the effectiveness of hierarchical disentanglement for a specific effect node, as determined by the explored causal DAG; 3) the method's ability to accurately identify the underlying DAG structure through exploration. A comprehensive demonstration of the conducted experiments is available online[2]. However, it is important to highlight two primary limitations of the experiments, which are detailed as follows:

Firstly, as an initial realization of the *relation-first* paradigm, RIRL struggles with modeling efficiency, since it requires a substantial amount of data points for each micro-causal relationship, making the heuristic exploration process slow. The dataset used is generated synthetically, thus providing adequate instances. However, current general-use simulation systems typically employ a single timeline to generate time sequences - It means that interactions of dynamics across multiple timelines cannot be showcased. Ideally, real-world data like clinical records would be preferable for validating the macro-causal model's generalizability. Due to practical constraints, we are unable to access such data for this study and, therefore, designate it as an area for future work. The issues of generalization inherent in such data have been experimentally confirmed in prior work Li et al. (2020), which readers may find informative.

Secondly, the time windows for the cause and effect, denoted by $n$ and $m$, were fixed at 10 and 1, respectively. This arose from an initial oversight in the experimental design stage, wherein the pivotal role of dynamic outcomes was not fully recognized, and our vision was limited by the RNN pattern. While the model can adeptly capture single-hop micro-causality, it struggles with multi-hop routines like $\mathcal{X} \to \mathcal{Y} \to \mathcal{Z}$, since the dynamics in $\mathcal{Y}$ have been discredited by $m = 1$. However, it does not pose a significant technical challenge to expand the time window in future works.

### 5.1 Hydrology Dataset

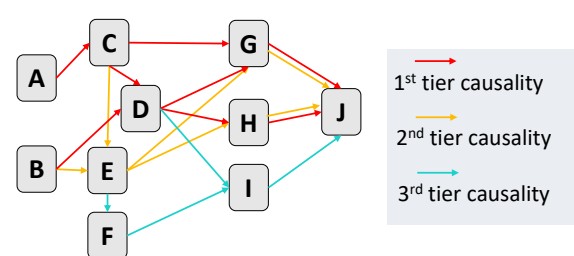

| ID | Variable Name | Explanation |
|----|---------------|-------------|
| A | Environmental set I | Wind Speed, Humidity, Temperature |
| B | Environmental set II | Temperature, Solar Radiation, Precipitation |
| C | Evapotranspiration | Evaporation and transpiration |
| D | Snowpack | The winter frozen water in the ice form |
| E | Soil Water | Soil moisture in vadose zone |
| F | Aquifer | Groundwater storage |
| G | Surface Runoff | Flowing water over the land surface |
| H | Lateral | Vadose zone flow |
| I | Baseflow | Groundwater discharge |
| J | Streamflow | Sensors recorded outputs |

Figure 17: Hydrological causal DAG: routine tiers organized by descending causality strength.

The employed dataset is from a widely-used synthetic resource in the field of hydrology, aimed at enhancing streamflow predictions based on observed environmental conditions such as temperature and precipitation. In hydrology, deep learning, particularly RNN models, has gained favor for extracting observational representations and predicting streamflow Goodwell et al. (2020); Kratzert et al. (2018). We focus on a simulation of the Root River Headwater watershed in Southeast Minnesota, covering 60 consecutive virtual years with daily updates. The simulated data is from the Soil and Water Assessment Tool (SWAT), a comprehensive system grounded in physical modules, to generate dynamically significant hydrological time series.

Figure 17 displays the causal DAG employed by SWAT, complete with node descriptions. The hydrological routines are color-coded based on their contribution to output streamflow: Surface runoff (the 1st tier) significantly impacts rapid streamflow peaks, followed by lateral flow (the 2nd tier); baseflow dynamics (the 3rd tier) have a subtler influence. Our exploration process aims to reveal these underlying tiers.

---

[2]https://github.com/kflijia/bijective_crossing_functions.git

Table 1: Characteristics of observables, and corresponding reconstruction performances.

| Variable | Dim | Mean | Std | Min | Max | Non-Zero Rate% | RMSE on Scaled | RMSE on Unscaled | BCE of Mask |
|---|---|---|---|---|---|---|---|---|---|
| A | 5 | 1.8513 | 1.5496 | -3.3557 | 7.6809 | 87.54 | 0.093 | 0.871 | 0.095 |
| B | 4 | 0.7687 | 1.1353 | -3.3557 | 5.9710 | 64.52 | 0.076 | 0.678 | 1.132 |
| C | 2 | 1.0342 | 1.0025 | 0.0 | 6.2145 | 94.42 | 0.037 | 0.089 | 0.428 |
| D | 3 | 0.0458 | 0.2005 | 0.0 | 5.2434 | 11.40 | 0.015 | 0.679 | 0.445 |
| E | 2 | 3.1449 | 1.0000 | 0.0285 | 5.0916 | 100 | 0.058 | 3.343 | 0.643 |
| F | 4 | 0.3922 | 0.8962 | 0.0 | 8.6122 | 59.08 | 0.326 | 7.178 | 2.045 |
| G | 4 | 0.7180 | 1.1064 | 0.0 | 8.2551 | 47.87 | 0.045 | 0.81 | 1.327 |
| H | 4 | 0.7344 | 1.0193 | 0.0 | 7.6350 | 49.93 | 0.045 | 0.009 | 1.345 |
| I | 3 | 0.1432 | 0.6137 | 0.0 | 8.3880 | 21.66 | 0.035 | 0.009 | 1.672 |
| J | 1 | 0.0410 | 0.2000 | 0.0 | 7.8903 | 21.75 | 0.007 | 0.098 | 1.088 |

Table 2: The brief results from the RIRL exploration.

| Edge | A→C | B→D | C→D | C→G | D→G | G→J | D→H | H→J | B→E | E→G | E→H | C→E | E→F | F→I | I→J | D→I |
|---|---|---|---|---|---|---|---|---|---|---|---|---|---|---|---|---|
| KLD | 7.63 | 8.51 | 10.14 | 11.60 | 27.87 | 5.29 | 25.19 | 15.93 | 37.07 | 39.13 | 39.88 | 46.58 | 53.68 | 45.64 | 17.41 | 75.57 |
| Gain | 7.63 | 8.51 | 1.135 | 11.60 | 2.454 | 5.29 | 25.19 | 0.209 | 37.07 | -5.91 | -3.29 | 2.677 | 53.68 | 45.64 | 0.028 | 3.384 |

## 5.2 Higher-Dimensional Reconstruction

This test is based on ten observable nodes, each requiring an individual autoencoder for initialing its higher-dimensional representation. Table 1 lists the characteristics of these observables after being scaled (i.e., normalized), along with their autoencoders' reconstruction accuracies, assessed in the root mean square error (RMSE), where a lower RMSE indicates higher accuracy for both scaled and unscaled data.

The task is challenged by the limited dimensionalities of the ten observables - maxing out at just 5 and the target node, $J$, having just one attribute. To mitigate this, we duplicate the input vector to a consistent 12-length and add 12 dummy variables for months, resulting in a 24-dimensional input. A double-wise extension amplifies this to 576 dimensions, from which a 16-dimensional representation is extracted via the autoencoder. Another issue is the presence of meaningful zero-values, such as node $D$ (Snowpack in winter), which contributes numerous zeros in other seasons and is closely linked to node $E$ (Soil Water). We tackle this by adding non-zero indicator variables, called *masks*, evaluated via binary cross-entropy (BCE).

Despite challenges, RMSE values ranging from 0.01 to 0.09 indicate success, except for node $F$ (the Aquifer). Given that aquifer research is still emerging (i.e., the 3rd tier baseflow routine), it is likely that node $F$ in this synthetic dataset may better represent noise than meaningful data.

## 5.3 Hierarchical Disentanglement

Table 3 provides the performance of stacking relation-indexed representations. For each effect node, the accuracies of its micro-causal relationship reconstructions are listed, including the ones from each single cause node (e.g., $B \to D$ or $C \to D$), and also the one from combined causes (e.g., $BC \to D$). We call them "single-cause" and "full-cause" for clarity. We also list the performances of their initialized variable representations on the left side, to provide a comparative baseline. In micro-causal modeling, the effect node has two outputs with different data stream inputs. One is input from its own encoder (as in optimization step 2), and the other is from the cause-encoder, i.e., indexing through the relation (as in optimization step 1). Their performances are arranged in the middle part, and on the right side of this table, respectively.

The KLD metrics in Table 3 indicate the strength of learned causality, with a lower value signifying stronger. Due to the data including numerous meaningful zeros, we have an additional reconstruction for the binary outcome as "whether zero or not", named "mask" and evaluated in Binary Cross Entropy (BCE).

For example, node $J$'s minimal KLD values suggest a significant effect caused by nodes $G$ (Surface Runoff), $H$ (Lateral), and $I$ (Baseflow). In contrast, the high KLD values imply that predicting variable $I$ using $D$ and $F$ is challenging. For nodes $D$, $E$, and $J$, the "full-cause" are moderate compared to their "single-cause" scores, suggesting a lack of informative associations among the cause nodes. In contrast, for nodes $G$ and $H$, lower "full-cause" KLD values imply capturing meaningful associative effects through hierarchical stacking. The KLD metric also reveals the most contributive cause node to the effect node. For example, the proximity of the $C \to G$ strength to $CDE \to G$ suggests that $C$ is the primary contributor to this causal relationship.

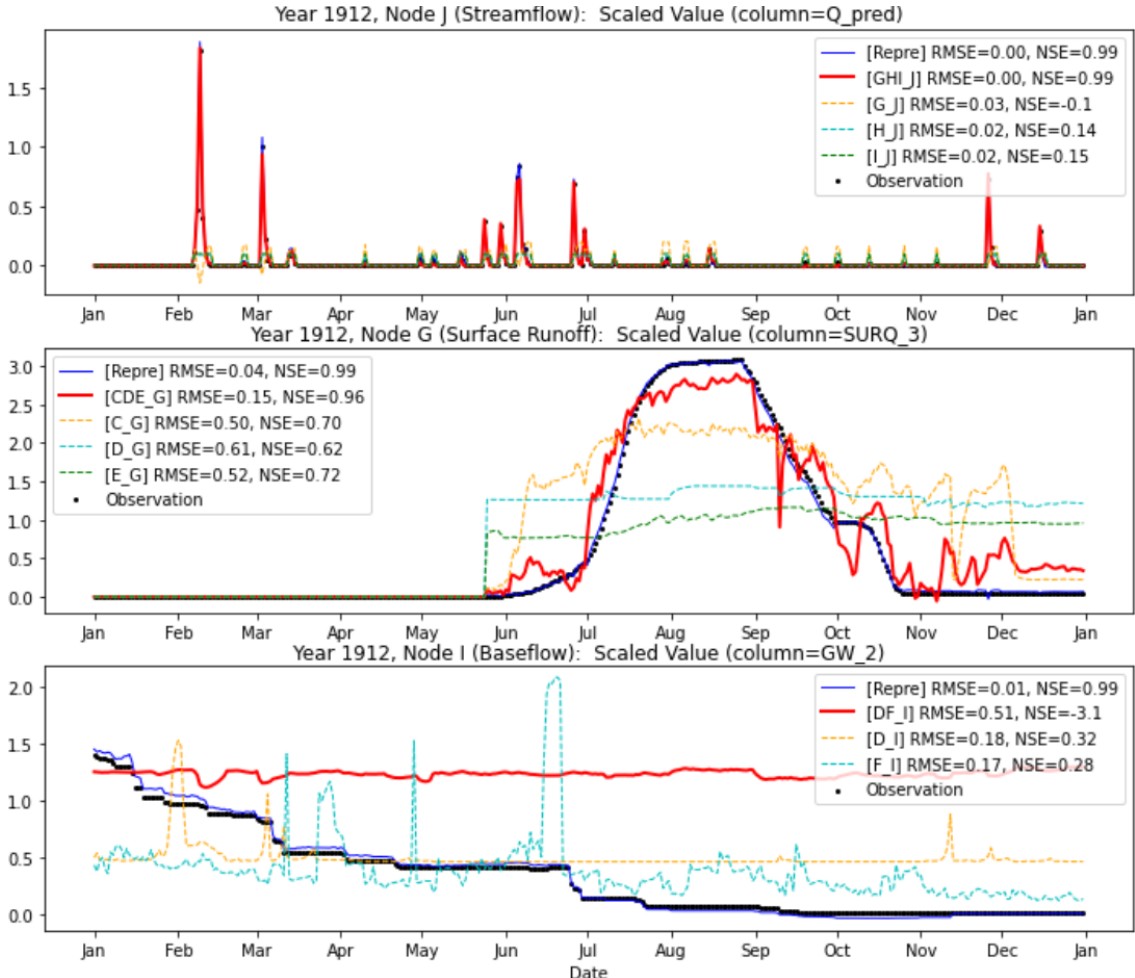

Figure 18: Reconstructed dynamics, via hierarchically stacked relation-indexed representations.

Figure 18 showcases reconstructed timing distributions for the effect nodes $J$, $G$, and $I$ in the same synthetic year to provide a straightforward overview of the hierarchical disentanglement performances. Here, black dots represent the ground truth; the blue line indicates the initialized variable representation and the "full-cause" representation generates the red line. In addition to RMSE, we also employ the Nash–Sutcliffe model efficiency coefficient (NSE) as an accuracy metric, commonly used in hydrological predictions. The NSE ranges from $-\infty$ to 1, with values closer to 1 indicating higher accuracy.

The initialized variable representation closely aligns with the ground truth, as shown in Figure 18, attesting to the efficacy of our proposed autoencoder architecture. As expected, the "full-cause" performs better than the "single-cause" for each effect node. Node $J$ exhibits the best prediction, whereas node $I$ presents a challenge. For node $G$, causality from $C$ proves to be significantly stronger than the other two, $D$ and $E$.

## 5.4  DAG Structure Exploration

The first round of detection starts from the source nodes $A$ and $B$ and proceeds to identify their potential edges, until culminating in the target node $J$. Candidate edges are selected based on their contributions to the overall KLD sum (less gain is better). Table 6 shows the detected order of the edges in Figure 17, accompanied by corresponding KLD sums in each round, and also the KLD gains after each edge is included. Color-coding in the cells corresponds to Figure 17, indicating tiers of causal routines. The arrangement underscores the effectiveness of this latent space exploration approach.

Table 4 in Appendix A displays the complete exploration results, with candidate edge evaluations in each round of detection. Meanwhile, to provide a clearer context about the dataset qualification with respect to underlying structure identification, we also employ the traditional causal discovery method, Fast Greedy

Table 3: Performances of micro-causal relationship reconstructions using RIRL, categorized by effect nodes.

| Efect Node | Variable Representation (Initialized) | | | Cause Node | Variable Representation (in Micro-Causal Models) | | | Relation-Indexed Representation | | | |
|---|---|---|---|---|---|---|---|---|---|---|---|
| | RMSE | | BCE | | RMSE | | BCE | RMSE | | BCE | KLD |
| | on Scaled Values | on Unscaled Values | Mask | | on Scaled Values | on Unscaled Values | Mask | on Scaled Values | on Unscaled Values | Mask | (in latent space) |
| C | 0.037 | 0.089 | 0.428 | A | 0.0295 | 0.0616 | 0.4278 | 0.1747 | 0.3334 | 0.4278 | 7.6353 |
| D | 0.015 | 0.679 | 0.445 | BC | 0.0350 | 1.0179 | 0.1355 | 0.0509 | 1.7059 | 0.1285 | 9.6502 |
| | | | | B | 0.0341 | 1.0361 | 0.1693 | 0.0516 | 1.7737 | 0.1925 | 8.5147 |
| | | | | C | 0.0331 | 0.9818 | 0.3404 | 0.0512 | 1.7265 | 0.3667 | 10.149 |
| E | 0.058 | 3.343 | 0.643 | BC | 0.4612 | 26.605 | 0.6427 | 0.7827 | 45.149 | 0.6427 | 39.750 |
| | | | | B | 0.6428 | 37.076 | 0.6427 | 0.8209 | 47.353 | 0.6427 | 37.072 |
| | | | | C | 0.5212 | 30.065 | 1.2854 | 0.7939 | 45.791 | 1.2854 | 46.587 |
| F | 0.326 | 7.178 | 2.045 | E | 0.4334 | 8.3807 | 3.0895 | 0.4509 | 5.9553 | 3.0895 | 53.680 |
| G | 0.045 | 0.81 | 1.327 | CDE | 0.0538 | 0.9598 | 0.0878 | 0.1719 | 3.5736 | 0.1340 | 8.1360 |
| | | | | C | 0.1057 | 1.4219 | 0.1078 | 0.2996 | 4.6278 | 0.1362 | 11.601 |
| | | | | D | 0.1773 | 3.6083 | 0.1842 | 0.4112 | 8.0841 | 0.2228 | 27.879 |
| | | | | E | 0.1949 | 4.7124 | 0.1482 | 0.5564 | 10.852 | 0.1877 | 39.133 |
| H | 0.045 | 0.009 | 1.345 | DE | 0.0889 | 0.0099 | 2.5980 | 0.3564 | 0.0096 | 2.5980 | 21.905 |
| | | | | D | 0.0878 | 0.0104 | 0.0911 | 0.4301 | 0.0095 | 0.0911 | 25.198 |
| | | | | E | 0.1162 | 0.0105 | 0.1482 | 0.5168 | 0.0097 | 3.8514 | 39.886 |
| I | 0.035 | 0.009 | 1.672 | DF | 0.0600 | 0.0103 | 3.4493 | 0.1158 | 0.0099 | 3.4493 | 49.033 |
| | | | | D | 0.1212 | 0.0108 | 3.0048 | 0.2073 | 0.0108 | 3.0048 | 75.577 |
| | | | | F | 0.0540 | 0.0102 | 3.4493 | 0.0948 | 0.0098 | 3.4493 | 45.648 |
| J | 0.007 | 0.098 | 1.088 | GHI | 0.0052 | 0.0742 | 0.2593 | 0.0090 | 0.1269 | 0.2937 | 5.5300 |
| | | | | G | 0.0077 | 0.1085 | 0.4009 | 0.0099 | 0.1390 | 0.4375 | 5.2924 |
| | | | | H | 0.0159 | 0.2239 | 0.4584 | 0.0393 | 0.5520 | 0.4938 | 15.930 |
| | | | | I | 0.0308 | 0.4328 | 0.3818 | 0.0397 | 0.5564 | 0.3954 | 17.410 |

Search (FGES), with a 10-fold cross-validation to perform the same procedure as RIRL exploration. The results in Table 5 are available in Appendix A, exhibiting the difficulties of using conventional methods.

## 6 Conclusions

This paper focuses on the inherent challenges of the traditional i.i.d.-based learning paradigm in addressing causal relationships. Conventionally, we construct statistical models as observers of the world, grounded in epistemology. However, adopting this perspective assumes that our observations accurately reflect the "reality" as we understand it, implying that seemingly objective models may actually be based on subjective assumptions. This fundamental issue has become increasingly evident in causality modeling, especially with the rise of applications in causal representation learning that aim to automate the specification of causal variables traditionally done manually.

Our understanding of causality is fundamentally based on the creator's perspective, as the "what...if" questions are only valid within the possible world we conceive in our consciousness. The advocated "perspective shift" represents a transformation from an *object-first* to a *relation-first* modeling paradigm, a change that transcends mere methodological or technical advancements. Indeed, this shift has been facilitated by the advent of AI, particularly through neural network-based representation learning, which lays the groundwork for implementing *relation-first* modeling in computer engineering.

The limitation of the observer's perspective in traditional causal inference prevents the capture of dynamic causal outcomes, namely, the nonlinear timing distributions across multiple "possible timelines". Accordingly, this oversight has led to compensatory efforts, such as the introduction of hidden confounders and the reliance on the sufficiency assumption. These theories have been instrumental in developing knowledge systems across various fields over the past decades. However, with the rapid advancement of AI techniques, the time has come to move beyond the conventional modeling paradigm toward the potential realization of AGI.

In this paper, we present *relation-first* principle and its corresponding modeling framework for structuralized causality representation learning, based on discussions about its philosophical and mathematical underpinnings. Adopting this new framework allows us to simplify or even bypass complex questions significantly. We also introduce the Relation-Indexed Representation Learning (RIRL) method as an initial application of the *relation-first* paradigm, supported by experiments that validate its efficacy.

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

# A    Appendix: Complete Experimental Results in DAG Structure Exploration Test

Table 4: The Complete Results of RIRL Exploration in the Latent Space. Each row stands for a round of detection, with '#' identifying the round number, and all candidate edges are listed with their KLD gains as below. 1) Green cells: the newly detected edges. 2) Red cells: the selected edge. 3) Blue cells: the trimmed edges accordingly.

| # | Candidate edges and KLD gains (in left→right order; (red)=selected, (green)=newly detected, (blue)=trimmed) |
|---|---|
| #1 | A→C 7.6354 (red); A→D 19.7407; A→E 60.1876; A→F 119.7730; B→C 8.4753 (blue); B→D 8.5147; B→E 65.9335; B→F 132.7717 |
| #2 | A→D 19.7407; A→E 60.1876; A→F 119.7730; B→D 8.5147 (red); B→E 65.9335; B→F 132.7717; C→D 10.1490 (green); C→E 46.5876; C→F 111.2978; C→G 11.6012; C→H 39.2361; C→I 95.1564 |
| #3 | A→D 9.7357 (red); A→E 60.1876; A→F 119.7730; B→E 65.9335; B→F 132.7717; C→D 1.1355 (red); C→E 46.5876; C→F 111.2978; C→G 11.6012; C→H 39.2361; C→I 95.1564; D→E 63.7348 (green); D→F 123.3203; D→G 27.8798; D→H 25.1988; D→I 75.5775 |
| #4 | A→E 60.1876; A→F 119.7730; B→E 65.9335; B→F 132.7717; C→E 46.5876; C→F 111.2978; C→G 11.6012; C→H 39.2361; C→I 95.1564; D→E 63.7348; D→F 123.3203; D→G 27.8798; D→H 25.1988; D→I 75.5775 |
| #5 | A→E 60.1876; A→F 119.7730; B→E 65.9335; B→F 132.7717; C→E 46.5876; C→G 11.6012 (red); C→H 39.2361; C→I 95.1564; D→E 63.7348; D→F 123.3203; D→G 27.8798; D→H 25.1988; D→I 75.5775; G→J 5.2924 (green) |
| #6 | A→E 60.1876; A→F 119.7730; B→E 65.9335; B→F 132.7717; C→E 46.5876; C→H 39.2361; C→I 95.1564; D→E 63.7348; D→F 123.3203; D→G 25.1988; D→H 25.1988; D→I 75.5775; G→J 5.2924 (red) |
| #7 | A→E 60.1876; A→F 119.7730; B→E 65.9335; B→F 132.7717; C→E 46.5876; C→H 39.2361 (blue); D→E 95.1564; D→F 63.7348; D→G 123.3203; D→H 25.1988 (red); D→I 123.3203 |
| #8 | A→E 60.1876; A→F 119.7730; B→E 65.9335; B→F 132.7717; C→E 46.5876; C→F 111.2978; C→I 95.1564; D→E 63.7348; D→F 123.3203; D→G 75.5775; D→I 75.5775; H→J 0.2092 (red) |
| #9 | A→E 60.1876 (blue); A→F 119.7730; B→E 65.9335; B→F 132.7717; C→E 46.5876 (blue); C→I 95.1564; D→E 63.7348; D→F 123.3203; D→F 123.3203; D→I 75.5775; E→H -3.2931; E→I 110.2558 |
| #10 | A→F 119.7730; B→E -6.8372 (red); B→F 132.7717; C→I 95.1564; D→E 123.3203; D→F 75.5775; D→I 75.5775; E→F 53.6806; E→G -5.9191 (green); E→H -3.2931; E→I 110.2558 |
| #11 | A→F 119.7730; B→F 132.7717; C→F 111.2978; C→I 95.1564; D→F 123.3203; D→I 75.5775; E→F 53.6806; E→G -5.9191 (red); E→H -3.2931 |
| #12 | A→F 119.7730; B→F 132.7717; C→F 111.2978; C→I 95.1564; D→F 123.3203; D→I 75.5775; E→F 53.6806; E→H -3.2931 (red) |
| #13 | A→F 119.7730 (blue); A→D 132.7717 (blue); A→E 111.2978 (blue); B→C 123.3203 (blue); B→D 17.0407 (blue); B→E 53.6806 (red); B→F 75.5775; C→I 95.1564; D→F 123.3203; E→F 75.5775; E→I 110.2558 |
| #14 | C→I 95.1564; D→I 75.5775; E→I 110.2558; F→I 45.6490 (red) |
| #15 | C→I 15.0222; D→I 3.3845; I→J 0.0284 (red) |
| #16 | C→I 15.0222 (blue); D→I 3.3845 (red) |

Table 5: Average performance of 10-Fold FGES (Fast Greedy Equivalence Search) causal discovery, with the prior knowledge that each node can only cause the other nodes with the same or greater depth with it. An edge means connecting two attributes from two different nodes, respectively. Thus, the number of possible edges between two nodes is the multiplication of the numbers of their attributes, i.e., the lengths of their data vectors. (All experiments are performed with 6 different Independent-Test kernels, including chi-square-test, d-sep-test, prob-test, disc-bic-test, fisher-z-test, mvplr-test. But their results turn out to be identical.)

| Cause Node | A | B | | C | | | D | | | E | | | F | G | H | I |
|---|---|---|---|---|---|---|---|---|---|---|---|---|---|---|---|---|
| True Causation | A→C | B→D | B→E | C→D | C→E | C→G | D→G | D→H | D→I | E→F | E→G | E→H | F→I | G→J | H→J | I→J |
| Number of Edges | 16 | 24 | 16 | 6 | 4 | 8 | 12 | 12 | 9 | 8 | 8 | 8 | 12 | 4 | 4 | 3 |
| Probability of Missing | 0.038889 | 0.125 | 0.125 | 0.062 | 0.06875 | 0.039286 | 0.069048 | 0.2 | 0.142857 | 0.3 | 0.003571 | 0.2 | 0.142857 | 0.0 | 0.072727 | 0.030303 |
| Wrong Causation | | | | C→F | | | | D→E | D→F | | | | F→G | G→H | G→I | H→I |
| Times of Wrongly Discovered | | | | 5.6 | | | | 1.2 | 0.8 | | | | 5.0 | 8.2 | 3.0 | 2.8 |

Table 6: Brief Results of the Heuristic Causal Discovery in latent space, identical with Table 3 in the paper body, for better comparison to the traditional FGES methods results on this page.
The edges are arranged in detected order (from left to right) and their measured causal strengths in each step are shown below correspondingly. Causal strength is measured by KLD values (less is stronger). Each round of detection is pursuing the least KLD gain globally. All evaluations are in 4-Fold validation average values. Different colors represent the ground truth causality strength tiers (referred to the Figure 10 in the paper body).

| Causation | A→C | B→D | C→D | C→G | D→G | G→J | D→H | H→J | C→E | B→E | E→G | E→H | E→F | F→I | I→J | D→I |
|---|---|---|---|---|---|---|---|---|---|---|---|---|---|---|---|---|
| KLD | 7.63 | 8.51 | 10.14 | 11.60 | 27.87 | 5.29 | 25.19 | 15.93 | 46.58 | 65.93 | 39.13 | 39.88 | 53.68 | 45.64 | 17.41 | 75.57 |
| Gain | 7.63 | 8.51 | 1.135 | 11.60 | 2.454 | 5.29 | 25.19 | 0.209 | 46.58 | -6.84 | -5.91 | -3.29 | 53.68 | 45.64 | 0.028 | 3.384 |

