# OpenReview forum: "Relation-First Modeling Paradigm for Causal Representation Learning toward the Development of AGI"
_TMLR — Rejected by TMLR_

### Review · Reviewer_TYVJ · 2023-12-08

**Summary Of Contributions:**

To provide context, I am a reviewer familiar with causal inference; broadly familiar with the literature on causal representation learning; somewhat familiar with the literature on causal learning in relational models.

The paper discusses two paradigms it calls “Observation-Oriented” and “Relation-Oriented” modeling, with the attempt to discuss how causal reasoning techniques could be incorporated into artificial intelligence systems.

**Audience:**

No

**Broader Impact Concerns:**

No concerns.

**Claims And Evidence:**

No

**Requested Changes:**

Therefore, in summary, I believe the authors should consider to (i) provide a clear problem statement of what they are attempting to demonstrate, in a scientific language with a high degree of mathematical rigor; (ii) explain all the necessary building blocks through example(s) in the very beginning; (iii) outline the key formal results (Theorems, Algorithms, Propositions) that are the result of their work. Without these, it is very difficult to appreciate what the contributions of the paper are.

**Strengths And Weaknesses:**

There are a number of major concerns about the papers contents:

1. The framing of the paper is extremely difficult to follow. For instance, the majority of the notation is not introduced in a clean way. Even in Section 1.1., the notation is unclear. For instance, how should one think about the relationship function f(X, \theta)? Could the authors introduce a clean example in which the basic ingredients and building blocks are explained? In Definition 1, for instance, some concepts are introduced in words, but their formal mathematical meaning is unclear.

2. In the same section, following Definition 1, there statements supposedly relating to introduced concepts to the notions from the causal representation literature. However, it is remains unclear what the formal connections here are. More broadly, is the notation borrowed from somewhere? I am unsure if there is a reference to some canonical text that includes all the required concepts. In the literature on relational models, concepts such as entity, relationships, attributes etc. I do not think any of these concepts are introduced adequately.

3.  The contributions and the aims of the paper are unclear: the paper's outline (consisting of four parts) does not mention any formal results. Are there any key Theorems that the paper has? Are there any important Propositions? Are there Algorithms that the paper develops? It seems that the paper is mostly introducing Definitions and providing Remarks, while the level of mathematical rigor is not very high for these.

4. It would be very beneficial to have a worked example early on in the paper, i.e., in the Introduction. Through the example, the authors could illustrate exactly what their aim is, and can ground the very abstract discussion that is taking place.

5. The scope of the paper seems to be remarkably broad. The paper seems to discuss AGI, LLMs, Causal Inference, and other topics as well. I believe the massive scope also dilutes the clarity and the focus of the paper.

---

### Review · Reviewer_wRNQ · 2023-12-22

**Summary Of Contributions:**

This paper addresses the problem of learning causal relationships from data. The authors key claim is that much of causal relationships that humans/agents use to reason is latent (matching observations from the cognitive science community), and that causal knowledge should accordingly be represented as such. The authors detail a framework which models the functional dependence between time series, and propose a learning framework that consists of learning a latent space representation, inferring the causal graphical model within said latent space, and then performing causal reasoning with respect to the latent space representations. A procedure is proposed which uses invertible autoencoders to provide a latent space representation. A set of evaluations are provided on a synthetic dataset from the hydrology literature.

**Audience:**

Yes

**Broader Impact Concerns:**

AGI in general has a bevy of larger impact concerns, which are entirely unaddressed by this paper. I actually believe these claims are out of scope, and the authors would do well from removing AGI as a motivating factor in the present work.

**Claims And Evidence:**

No

**Requested Changes:**

It’s not clear how the definition of Relation-Induced representation (definition 1) would vary from what we encounter within the causal graphical models literature (or even the non-causal graphical models). For example, if we think of the definition given by Pearl (Causality, 1999), we assume some functional parameterization exists which accounts for the components of $\mathcal{Y}$ which are explainable via $\mathcal{X}$. I suppose the key differentiation is that the authors are considering the every time step has been observed and are employing a vector based representation, but the delineation isn’t entirely clear to me. The key piece here is that you are assuming some functional graphical model where we have vector valued representation? How is this similar / different from prior functional representations (e.g., Zhu et al (2016)?

I’m not sure why this is being referred to as “relation-oriented”, the definition of relation and it’s distinguishment from dependence is not clear to me. Can I just replace “relationship model” with “dependence model”? If this is entirely with respect to probabilistic relationships, how is this distinguished from causal models which deal with nonparametric causal relationships via dependence (e.g., CGMs, ADMGS, PAGS/MAGS).

It would be interesting if the authors could compare to the work on macro causal modeling from Eberhardt, which seems to be related in that both lines of work are reasoning over causal relationships on multiple scales.

I don’t entirely follow definition 6 here. Correlation has a precise statistical meaning which is not a directed relationship, and does not require “static cause and effect”. In fact there is no reason at all to assume a causal relationship in the presence of statistical correlation.

In the definition of the interventional effect before the beginning of 3.2, is this assuming a binary $x_t$ and keeping it fixed to the same value throughout the entirety of the time series? We can also easily define estimands that fix $x$ at a specific time point $t=i$ and then allow it to take on arbitrary values afterwards. Also, we need not restrict to a binary variable. It would be useful if the authors could clarify / update some of this definition.

I’m a little confused by the points made in 4.2. Regression adjustment / IPW / AIPW / etc. allow someone to estimate the direct effect after identification. There is a vast literature (too vast to list here) developing these methods for accounting for nonlinear relationships using semiparametric. Further, it’s not entirely clear what the proposed alternative is. It would seem that it is to have a fully parametrized graphical model/parameterized graphical model in latent space. However, this problem is strictly more difficult than the problem of identification of a single estimand.

It seems like the authors crucial point here is not so much about relationships but rather latent representation learning implied by modeling dependencies/relationships between observations. The authors develop a hierarchy, which I think is interesting. However, the solution provided here (composition of autoencoders), while intriguing do not seem to have any guarantees, other than intuition motivated by the prior sections, on why we should believe that this approach will recover representations that result in more robust generalization behavior. Assuming a sufficient latent space representation exists also exists in other causal contexts (e.g., Veitch, et al. (2019)). The key difficulty is in providing a learning procedure that provides a set of necessary and sufficient representations for general causal reasoning. I think it’s fine to establish a framework and then explicitly appeal to heuristics / constraining assumptions on the representation learning procedure but I think it’s important to make that plain in the text. My key concern here is that the authors seem to stack notoriously difficult problems together (representation learning and causal discovery) and that a small error in either could cascade and lead to erroneous results. I would be interested to hear how the authors think about this issue / general guidance for practice.

Finally, the evaluation provided doesn’t quite measure the claims made throughout the paper that the key goal is in improving generalization behavior in the pursuit of “AGI”. My suggestion overall here would be to edit the text substantially to revise much of the claims and aims to be more targeted and to avoid large claims such as ‘toward agi’. This paper has valuable insights but I think some of them are lost by trying to tie the proposed methods and observations past the scope of what is addressed in the methodology.


Zhu, Hongxiao, Nate Strawn, and David B. Dunson. "Bayesian graphical models for multivariate functional data." (2016).

Veitch, Victor, Yixin Wang, and David Blei. "Using embeddings to correct for unobserved confounding in networks." Advances in Neural Information Processing Systems 32 (2019).

**Strengths And Weaknesses:**

Overall, I think this is an interesting idea but much of the text could be improved. As I list below in more detail, the authors have a narrative that is a little sprawling, and it can be difficult to follow the thread of what precisely is being proposed and in some places there is a lack of precision in terms of claims.

---

> ### Author Response · Authors · 2024-02-16
>
> Dear Reviewer wRNQ,
>
> I would like to extend my heartfelt thanks to you for providing critical guidance and references that point me toward the relevant philosophical literature.
>
> Given my advisor's antipathy towards philosophical discussions, I had grown accustomed to steering clear of philosophy, which kept me at a distance from truly clarifying my theory and led to chaotic presentations. Your insights were invaluable to me.
>
> The current version has been significantly improved and streamlined.
>
> Looking forward to your feedback, and thanks again!

---

### Review · Reviewer_baSu · 2024-01-19

**Summary Of Contributions:**

The contribution of the paper is unclear. It is not easy to identify the goal of the paper

**Audience:**

No

**Claims And Evidence:**

No

**Requested Changes:**

Focus on the objectives and the results

**Strengths And Weaknesses:**

The paper should be reduced in order to stress its focus and the corresponding results

---

> ### Author Response · Authors · 2024-02-16
>
> Dear Reviewer baSu,
>
> I concur with your concise evaluation of the previous version. Over the past month, I have diligently considered your suggestions while revising the current version, resulting in significant clarification.
>
> Looking forward to your feedback and apologize for the month-long wait!

---

### Author Response · Authors · 2023-12-20
**Hoping for quick feedback and early holiday wishes**

Dear reviewers and editors,

I sincerely appreciate your commitment to reviewing my paper, especially given its foundational topic and extensive length.

The timeliness of your feedback is crucial for my ability to legally remain in this country and embark on my research career.
I would be immensely grateful for any early feedback you might provide, though I intend no pressure by this request.

Wishing you and your family a Merry Christmas in advance!

Best

---

### Decision · Action_Editor_SzN4 · 2024-04-19

**Recommendation:** Reject

**Comment:**

Private discussion among the reviewers echoed the sentiment expressed in the original reviews and subsequent discussion with the author. There is clear consensus among all reviewers that the paper requires substantial revision before it is ready to be published. Detailed suggestions are included in the individual reviews.

**Audience:**

Yes, but the scope would need to be narrowed significantly and its contributions clarified.

**Claims And Evidence:**

Although the article presents an innovative approach to learning causal relationships from data, there is clear consensus among the reviewers that the paper requires substantial revision from its revised form and is not yet ready for publication. The main concerns revolve around the clarity and precision of the paper’s narrative and lack of clear contributions supported by evidence. The reviewers have provided extensive suggestions for improvement which have not been adequately met in the subsequent revisions.

Further, the paper is rather ambitious in its connections to AGI, with the reviewers suggesting that the paper is rather overreaching regarding AGI given its current content and would benefit from significantly toning down or eliminating these connections. Narrowing the scope of the paper accordingly and more precisely aligning the experimental evidence to the claims would help solidify the paper’s contributions.

Overall, the reviewer’s recognized the paper for its novel ideas, but it needs significant improvement in presentation, clarity, narrative cohesion, and empirical validation to meet the expectations of TMLR.